# Yeast as a Model to Understand Actin-Mediated Cellular Functions in Mammals—Illustrated with Four Actin Cytoskeleton Proteins

**DOI:** 10.3390/cells9030672

**Published:** 2020-03-10

**Authors:** Zain Akram, Ishtiaq Ahmed, Heike Mack, Ramandeep Kaur, Richard C. Silva, Beatriz A. Castilho, Sylvie Friant, Evelyn Sattlegger, Alan L. Munn

**Affiliations:** 1School of Medical Science, Gold Coast campus, Griffith University, Southport, QLD 4222, Australia; zain.akram@griffithuni.edu.au (Z.A.); ishtiaq.ahmed@griffithuni.edu.au (I.A.); heike.mack@griffithuni.edu.au (H.M.); 2Université de Strasbourg, CNRS, Génétique Moléculaire Génomique Microbiologie GMGM UMR 7156, F-67000 Strasbourg, France; waraichramandeep21@gmail.com (R.K.); s.friant@unistra.fr (S.F.); 3Department of Mechanistic Cell Biology Max-Planck Institute of Molecular Physiology, Dortmund 44227, Germany; richard.cardosodasilva@mpi-dortmund.mpg.de; 4Department of Microbiology, Immunology and Parasitology, Escola Paulista de Medicina, Universidade Federal de São Paulo, São Paulo 04023-062, Brazil; bacastilho@unifesp.br; 5School of Natural and Computational Sciences, Massey University, P.O. Box 102 904, North Shore Mail Centre, Albany, Auckland 0745, New Zealand; e.sattlegger@massey.ac.nz

**Keywords:** BAR domain, cancer, cytokinesis, endocytosis, F-BAR domain, Myc, translation factors, tumor suppressor, WASP, Wiskott-Aldrich Syndrome

## Abstract

The budding yeast *Saccharomyces cerevisiae* has an actin cytoskeleton that comprises a set of protein components analogous to those found in the actin cytoskeletons of higher eukaryotes. Furthermore, the actin cytoskeletons of *S. cerevisiae* and of higher eukaryotes have some similar physiological roles. The genetic tractability of budding yeast and the availability of a stable haploid cell type facilitates the application of molecular genetic approaches to assign functions to the various actin cytoskeleton components. This has provided information that is in general complementary to that provided by studies of the equivalent proteins of higher eukaryotes and hence has enabled a more complete view of the role of these proteins. Several human functional homologues of yeast actin effectors are implicated in diseases. A better understanding of the molecular mechanisms underpinning the functions of these proteins is critical to develop improved therapeutic strategies. In this article we chose as examples four evolutionarily conserved proteins that associate with the actin cytoskeleton: (1) yeast Hof1p/mammalian PSTPIP1, (2) yeast Rvs167p/mammalian BIN1, (3) yeast eEF1A/eEF1A1 and eEF1A2 and (4) yeast Yih1p/mammalian IMPACT. We compare the knowledge on the functions of these actin cytoskeleton-associated proteins that has arisen from studies of their homologues in yeast with information that has been obtained from in vivo studies using live animals or in vitro studies using cultured animal cell lines.

## 1. Introduction

### 1.1. Yeast as a Model Organism

Budding yeast (*Saccharomyces cerevisiae*) is a popular experimental model organism for the study of cellular processes. *S. cerevisiae* is unicellular and non-motile and because it is a eukaryote, it possesses a nucleus, mitochondria and both secretory and endocytic organelles like human cells. These membrane-bound organelles are easy to visualize by microscopy (e.g., vacuoles occupy 1/3–1/5 of the cell volume). *S. cerevisiae* is well-suited for live-cell imaging studies because it survives at room temperature and without a supply of growth factors, nutrients or carbon dioxide (unlike mammalian cells). *S. cerevisiae* is easy to culture in the laboratory (on rich media or chemically defined synthetic media), grows rapidly (doubling time of 90 min in rich media) at 30 °C, grows as an even cell suspension in liquid culture, forms discrete colonies on solid media, and its growth media are relatively inexpensive compared to those required by animal cells. This makes it easy and economical to obtain a large mass of yeast cells for use in biochemical approaches, e.g., subcellular fractionation, enzyme purification, isolation of protein complexes, transcriptomics, lipidomics and proteomics, etc [1,2,3,4,5].

*S. cerevisiae* reproduces by budding, i.e., a daughter cell grows from a point on the surface of the mother cell. This makes it possible to identify the stage of the cell cycle based on the presence or absence of a visible bud and the size of the bud relative to the mother cell, e.g., G1 cells have no bud, S-phase cells have a small to medium-sized bud and G2 and M phase cells have a large bud (Figure 1). Unlike animal cells which must be oncogenically transformed in order to proliferate indefinitely in cell culture (and therefore exhibit altered cell cycle regulation), *S. cerevisiae* proliferates in cell culture while retaining normal cell cycle regulation). Other advantages of *S. cerevisiae* include the existence of both high- and low-copy-number plasmids that can be easily transformed into yeast, regulated promoters (e.g., galactose-inducible) and the fact that relatively few *S. cerevisiae* genes contain introns, so one can often use genomic DNA instead of cDNA for the purposes of gene cloning and protein expression [2,3,4].

*S. cerevisiae* can propagate indefinitely as either a diploid or a haploid cell type. Both have a similar cell morphology (diploid cells being larger than haploid cells). There are two haploid cell types, **a** and α, which can be mated to form diploids (**a**/α). Diploids can be induced to undergo meiosis to yield four recombinant spores: two **a** and two α haploids. Because these four spores are held together by what remains of the mother cell wall (known as an ascus) it is possible to use a microscope fitted with a micromanipulator to recover all four haploid products of a single meiosis. This makes budding yeast ideal for performing genetic crosses to demonstrate Mendelian inheritance of mutations affecting nuclear genes by the progeny. Moreover, non-Mendelian inheritance of phenotypes (e.g., cytoplasmic inheritance of mitochondrial genes, dsRNA viruses and prions) is easy to identify. That budding yeast can be propagated as a diploid allows for mutants (e.g., gene deletion strains) defective in essential genes to be maintained as heterozygous diploids. On the other hand, mutations in non-essential genes (including recessive mutations, e.g., gene deletions) can be phenotypically characterized in haploids. Due to its ability to take up exogenous DNA and its efficient homologous recombination system, budding yeast is easy to genetically manipulate (e.g., to knock out genes, knock-down gene expression, mutate genes, overexpress genes, tag genes with reporters or introduce new genes) [1,2,3,4,6].

The *S. cerevisiae* genome has been fully sequenced and indeed the genomes of many commonly used *S. cerevisiae* laboratory and industrial strains have been fully sequenced (*Saccharomyces* Genome Database, website http://www.yeastgenome.org) [1,7,8]. Of the approximately 6000 genes in *S. cerevisiae*, complete gene knock-out mutants have been constructed for most genes [9]. In the case of non-essential genes haploid strains in which these genes are deleted remain viable, however in the case of essential genes haploids in which these genes are deleted are inviable [9]. For most essential genes, mutant haploid strains in which these genes are under the control of titratable promoters have been constructed so expression of most essential genes can be experimentally knocked-down, e.g., by addition of doxycycline [6].

An extensive set of genome-wide tools have been developed for use with *S. cerevisiae*. As well as genome-wide collections of gene knock-out and regulated knock-down mutant strains there are now fluorescently-tagged versions of most *S. cerevisiae* gene products and these have been used to create a database of the subcellular localization patterns and protein abundance under different environmental conditions of most *S. cerevisiae* gene products [10,11]. There are also databases of genome-wide gene expression data that include data showing the effect of changing various environmental conditions on relative gene expression levels [12,13]. Moreover, databases are available on genome-wide genetic interactions (e.g., gene-gene phenotypic enhancement or suppression) [14] as well as genome-wide physical associations of gene products and formation of protein-protein, protein-DNA, and protein-small molecule complexes e.g., [15,16,17,18,19,20].

*S. cerevisiae* is well suited for use in identifying targets for drugs derived from natural products. The majority of widely used drugs are derivatives of natural products synthesized by various species of soil bacteria belonging to the genus *Streptomyces* [21,22]. These natural compounds are used by soil bacteria such as *Streptomyces* to inhibit the growth of fungi, with which they compete for nutrients in their natural environment. The pharmacological effects of these natural products and their derivatives on human cells result from the high degree of conservation of biological processes between single-cell and multiple-cell eukaryotes. As fungal (including yeast) proteins are the natural targets that these natural products have evolved to inhibit, *S. cerevisiae* is sensitive to a wide range of human therapeutics and is therefore ideal for use in identifying target pathways of uncharacterized natural products using chemical-genetic interactions. An extensive database of chemical-genomic interactions is now available for *S. cerevisiae* [22].

A major advantage of yeast as an experimental model organism is the vast published literature on *S. cerevisiae* genes and mutant phenotypes (dating back to the 1930s). There is more information available on the molecular mechanisms that underpin cellular processes in the yeast cell than perhaps any other cell type including human cells. The amino acid sequence and functional conservation between *S. cerevisiae* and human proteins means that important insights into human disease mechanisms can be obtained from yeast studies. Increasingly, research into human disease genes is being facilitated by findings obtained on the homologous *S. cerevisiae* proteins [23,24,25,26,27].

The ability to combine classical genetic, molecular biology, biochemical and cell biology approaches using the same organism (as described above) as well as the existence of an actin cytoskeleton with components conserved between yeast and humans have made *S. cerevisiae* a particularly good experimental model for study of the actin cytoskeleton and actin-dependent cellular processes [28,29,30,31,32,33,34,35].

Additionally, whereas humans have multiple isoforms of each actin cytoskeleton protein, encoded in many cases by distinct genes, the actin cytoskeleton proteins in *S. cerevisiae* are often encoded by single genes (e.g., actin is encoded by a single gene in *S. cerevisiae* but by 6 genes in humans) [32,36,37,38,39]. Because there is often only one gene for an actin cytoskeletal protein of interest, deletion of a single gene in *S. cerevisiae* is often sufficient to confer a phenotype and this in turn enables the elucidation of the role of that actin cytoskeleton protein. *S. cerevisiae*, being unicellular, also simplifies the phenotypic characterization of mutations affecting actin cytoskeleton components compared to metazoans where the effect of gene disruption on various different cell and tissue types (e.g., blood cells, cardiac and skeletal muscle) would need to be investigated. Finally, yeast cells are quite robust and survive loss of many actin cytoskeleton proteins that in mammals are essential for life [40].

In this article we discuss how yeast studies help in understanding the function of evolutionarily conserved proteins associated with the actin cytoskeleton, using four proteins as examples: (1) yeast Hof1p/mammalian PSTPIP1, (2) yeast Rvs167p/mammalian BIN1, (3) yeast eEF1A/eEF1A1 and eEF1A2 and (4) yeast Yih1p/mammalian IMPACT (Table 1). Before discussing these actin cytoskeleton proteins in detail, it is necessary to first give a brief introduction to the yeast actin cytoskeleton and review its structure, assembly and cellular roles.

### 1.2. Actin Cytoskeleton in Yeast

The actin cytoskeleton plays a central role in governing the morphogenetic alterations that accompany cell division in all eukaryotic cells, including those of budding yeast. Rearrangement of actin networks also regulates other vital processes such as endocytosis and cell polarization. These rearrangements are regulated by 20 to 30 highly conserved actin-associated proteins. The budding yeast *S. cerevisiae* has been widely used as a model organism to study the actin cytoskeleton [31,32].

The yeast actin cytoskeleton comprises three distinct filamentous (F-) actin-rich structures: cortical actin patches (or spots), cytoplasmic actin cables and a contractile actomyosin ring [31,32,35,36,37,41,42,43,44,45,46,47,48,49,50,51,52]. Cortical actin patches are small (200 nm diameter) spots found at the cell cortex and are highly motile. Cytoplasmic actin cables are long and thin and extend through the cortical cytoplasm. Cytoplasmic actin cables function to direct the traffic of organelles [53,54,55], secretory vesicles [35,55,56] and mRNAs [35,55,57,58] to sites of growth [59,60]. While cortical actin patches and cytoplasmic actin cables are present in yeast cells in every stage of the cell division cycle [31,32,36,37,61,62], the contractile actomyosin ring forms only in dividing (mitotic) yeast cells and persists only until the completion of cytokinesis [35,41,42,43,44,45,46,47,48,49,50,51,52]. The distribution of cortical actin patches and cytoplasmic actin cables within yeast cells is polarized to sites of bud growth and cell division [31,32,36,37,55] (Figure 1). Cortical actin patches are often observed at the tips of cytoplasmic actin cables at the cortex of the bud [61,62] Cortical actin patches represent sites of endocytosis [31,32,33,63,64]. Due to their association with the tips of cytoplasmic actin cables, cortical actin patches may also contribute to polarized exocytosis as exocytic vesicles are transported via actin cables to the cortex [55,56,65]. Early in the cell cycle (G1) both cortical actin patches and cytoplasmic actin cables are polarized towards the nascent bud site. Subsequently, during bud emergence (S) they polarize to the growing small bud. When the bud approaches the size of the mother cell, both cortical actin patches and cytoplasmic actin cables redistribute to the medial region (neck) between the large bud and the mother cell [31,32,36,37,55]. The contractile actomyosin ring, on the other hand, forms only at the medial region and contracts from a ring to a point coincident with cytokinesis and then disappears (Figure 1) [35,41,42,43,44,45,46,47,48,49,50,52].

Although cortical actin patches, cytoplasmic actin cables and the contractile actomyosin ring all consist of F-actin, the type of F-actin differs between the various structures. Cortical actin patches consist of a branched (dendritic) network of short actin filaments [66,67]. The assembly of branched actin filaments is nucleated by a seven-subunit protein complex whose two largest subunits (Arp2p and Arp3p) are actin-related proteins known as the Arp2/3 complex [68]. The Arp2/3 complex binds to the sides of existing actin filaments and the Arp2p and Arp3p subunits function as a template for assembly of short daughter filaments that are oriented at 70° with respect to the mother actin filament [68]. Cytoplasmic actin cables and the contractile actomyosin ring both consist of linear (i.e., non-branched) actin filaments. Linear actin filaments are assembled, not by the Arp2/3 complex, but rather by another class of actin filament nucleator known as the formins. The formins accept actin monomers bound to either of two actin-monomer-binding proteins, profilin or Bud6p (also known as Aip3p) for filament assembly [51,69,70,71,72,73,74,75]. Many actin cytoskeleton proteins are found in both cortical actin patches and in cytoplasmic actin cables, the exception being the tropomyosins Tpm1p and Tpm2p which are found in actin cables, but not in cortical actin patches [56]. Formins have a conserved function in cell polarity [76].

### 1.3. Polarization of the Actin Cytoskeleton during the Budding Cycle of Yeast

Polarization of the *S. cerevisiae* actin cytoskeleton in late G1 to a site at the cortex where the daughter cell (bud) will form is under the overall control of two regulatory Rho-family GTPases: Rho1p and Cdc42p. In its GTP-bound form, Rho1p plays a role in polarized localization of Cdc42p to the nascent bud site and this role appears distinct from its role in activation of β-glucan synthases (see below) [77,78]. Next, in its GTP-bound form Cdc42p interacts with a set of downstream effector proteins that initiate recruitment of various proteins to the nascent bud site. A complex of proteins known as the polarisome then forms at the site on the cortex where the nascent bud will form. Polarisome components include: Spa2p, the Spa2p homologue Sph1p, the formin Bni1p, the actin-monomer binding protein Bud6p, Pea2p, Msb3p and Msb4p [79,80,81]. GTP-bound Cdc42p results in recruitment and activation of Bni1p (via the Cdc42p effector Gic2p) [82,83,84]. Activated Bni1p is responsible for the nucleation step that initiates the assembly of the linear actin filaments that comprise a subset of actin cables that are attached to the nascent bud site at the cortex and later, when the bud forms, throughout the cortex of the growing bud (assembly of another subset of actin cables by Bnr1p will be described later) [35,59]. The actin filament assembly activity of Bni1p is directly regulated by GTP-bound (i.e., activated) Rho GTPases, in particular Rho1p [35,70,82]. Bud6p is believed to supply the actin monomers for formin-dependent actin filament nucleation [71,72]. The cytoplasmic actin cables assembled by Bni1p direct secretory vesicles containing newly-synthesized membrane and cell wall material to the nascent bud site and then to a number of distinct sites within the growing bud [35,56,59].

Another formin, Bnr1p, localizes after the bud has formed and in contrast to Bni1p it localizes to the bud neck (a constriction between mother cell and bud) [35,59,60,75,85,86]. Bnr1p also assembles linear actin filaments but these are used to assemble a distinct subset of actin cables that initiate at the bud neck and extend into the mother cell [35,59]. This subset of cytoplasmic actin cables directs transport from the mother cell to the neck [35,59,60].

### 1.4. Role of Septins in Defining the Nascent Bud Site and Bud Neck in Yeast

A set of proteins that also localizes very early to the nascent bud site in *S. cerevisiae* comprises the mitotic septins (Cdc3p, Cdc10p, Cdc11p, Cdc12p and Shs1p/Sep7p) [35,87,88,89,90,91]. The septins are GTP-binding proteins that form hetero-octameric rod-shaped complexes which then assemble to form long (32–100 nm) membrane-associated filaments that are 7–9 nm in diameter [35,88,89,90,91,92]. These septin filaments assemble into a small ring that surrounds the nascent bud site [35,87,90,91,93,94]. They remain at the bud neck throughout the cell cycle, but undergo rearrangements/remodelling to form different structures at different stages of the cell cycle [35,87,90,91,94]. The septin ring has been proposed to serve as a diffusion barrier so membrane proteins and other cortical proteins cannot freely diffuse out of the bud neck region or bud into the mother cell [35,48,89,91,95,96].

As well as its role in actin filament assembly, the formin Bni1p (as well as Cdc42p and its GTPase-activating proteins/GAPs, polarisome components, protein kinases Cla4p and Gin4p and F-actin) also play a critical role in the recruitment of septins to the site of septin ring assembly at the cortex and/or in the assembly of the septin filaments at the cortex to form a septin ring [35,90,91,93,94,97]. The requirement for proteins like Bni1p and F-actin (which are essential for actin cable assembly) and the polarisome in septin ring assembly may reflect a role for polarized actin-cable-dependent delivery of a septin ring assembly factor whose distribution is restricted by the polarisome to a polarized cortical site. In contrast to Bni1p, which plays a role in septin ring assembly, the bud neck localization of the other formin, Bnr1p, is dependent on the septins and Bnr1p functions at a later cell cycle stage, v.i.z. cytokinesis (to be discussed later) [35,86].

### 1.5. Formation of Cortical Actin Patches and Their Function in Endocytosis

Cortical actin patches are visible throughout the cell division cycle [4,32,36,37]. However, individual cortical actin patches have a short lifespan (~7–15 s) and are highly motile [53,63,64,98,99,100,101,102]. The precursors of cortical patches continuously form on the plasma membrane at specific cortical sites, initiate the assembly of F-actin, undergo short-range movement from the cortex into the cell body accompanied by ongoing F-actin assembly, detach from the plasma membrane, undergo long-range movements accompanied by further F-actin assembly before finally dissociating in the cytoplasm [31,33,53,64,101,102]. The polarized distribution of cortical actin patches reflects a polarized distribution of the sites of cortical actin patch assembly [100]. A key study showed that the polarization of cortical actin patches is dependent on a polarized orientation of cytoplasmic actin cables (i.e., polarized along the mother cell-bud axis) whereas, in contrast, cytoplasmic actin cable polarization is not dependent on a polarized distribution of cortical actin patches [56]. Cortical actin patches are sites of endocytosis and internalize cargo membrane proteins and fluorescent water-soluble and lipid-soluble dyes [31,33,53,63,64,101,102].

The assembly of a cortical actin patch is initiated by the recruitment of the endocytic machinery, e.g., Ede1p, the endocytic vesicle coat protein clathrin (comprising the clathrin heavy chain Chc1p and light chain Clc1p), and the Fes/CIP4 Homology (FCH)-BIN1/Amphiphysin/Rvs (BAR) (F-BAR) domain-containing protein known as Syp1p. Other components of the endocytic machinery are then recruited to the nascent endocytic sites in a strict temporal sequence, including the early coat proteins, such as endocytic vesicle coat adaptor protein/clathrin assembly protein 2 (AP-2) (comprising Apl1p, Apm4p, Apl3p, Aps2p) and AP-180 (comprising Yap1801p and Yap1802p), followed by the mid-coat proteins, such as Sla2p, Ent1p, Ent2p, and then by the late coat proteins, such as Pan1p, Sla1p and End3p [64,101]. The coat proteins include adaptor molecules that recognize and bind to sorting signals present in the cytoplasmic tails of various endocytic cargo membrane proteins. As a consequence, the endocytic membrane cargo proteins and endocytic coat proteins cluster at the nascent endocytic sites [101]. Concomitant with the recruitment of the late coat proteins, the key actin filament assembly regulator proteins Las17p, the unconventional type I myosin (Myo5p) (and its partner protein Vrp1p) are recruited to the nascent endocytic site [101,102]. The recruitment of actin filament assembly regulators is significant because although the branched (dendritic) actin filaments that form on the surface of cortical actin patches are nucleated by the Arp2/3 complex (see above), Arp2/3 in both yeast and mammals requires the binding of activator proteins (known collectively as nucleation promoting factors or NPFs) for its activity. Yeast has several proteins with NPF activity, however, the most important NPFs for cortical actin patch assembly and endocytosis are Las17p and a complex of Vrp1p with the type I myosin Myo5p [102].

The recruitment of the first NPFs (e.g., Las17p) to the precursor of the cortical actin patch is followed temporally by the recruitment of actin, more NPFs (e.g., Myo5p) and then by recruitment of the Arp2/3 complex itself and the actin-binding protein Abp1p [63,102]. Then the precursor of the cortical actin patch assembles F-actin and becomes a mature cortical actin patch [63,102]. Live cell imaging studies have shown that precursors of cortical actin patches appear to be constrained because only random short-range movements occur [63,102]. This initial phase is followed by slow (~25 nm/s), short-range (~200 nm) movements directed away from the cortex and into the cell interior [63,64,102,103]. Following the recruitment of the amphiphysin homologs Rvs161p and Rvs167p [64,104] and the dynamin-related protein Vps1p [104], a membrane scission event is thought to occur and this corresponds to the time when cortical actin patches are observed to initiate fast (~230 nm/s) long-range (~500–1000 nm) movements into the cell interior [63]. It is believed that new F-actin assembly at the cortex and Myo5p motor activity both provide force to move the actin patch and its internalized endocytic cargo through the cytoplasm during directed short-range movement. This is based on the observation that Myo5p motor activity and nucleation promoting factor (NPF) activity are both required for inward movement [102]. Long-range movement of actin patches and associated endocytic vesicles through the cytoplasm is thought to be due in large part (although perhaps not only) to Myo5p motor-dependent retrograde actin filament flow away from the cortex into the cell body [63,64,102]. Interestingly, some actin patch proteins do not move from the cortex (e.g., Las17p, Myo5p), some undergo slow inward movement only (e.g., Sla1p, Sla2p, Pan1p) and others undergo fast long-range movement as well (e.g., Arc15p, Abp1p) [63,102]. Disassembly of actin patches after long-range movements has been proposed to occur once the actin patch interacts with endosomes [53]. Some studies find an association of actin patches with actin cables during long-range fast movement and this may play a role in such fast movement [53,61].

### 1.6. Overview of Cytokinesis in Yeast and Mammals

In both yeast and mammalian cells a contractile actomyosin ring plays a central role in cytokinesis [35,41,42,43,44,46,48,49,50,51,52,105]. Moreover, the contractile actomyosin rings of mammalian cells and yeast have a similar overall appearance and a similar protein composition indicating a high degree of evolutionary conservation [41,42,43,44,46,48,49,50,51,52,105].

However, some aspects of actomyosin ring assembly and function have diversified to meet the specific requirements of cytokinesis in mammalian cells and yeast cells [41,44,50,52,105]. First, budding yeast cells, being much smaller than mammalian cells (and also with a constricted bud neck), have an actomyosin ring with a smaller diameter (~1 μm compared to 10–30 μm in mammalian cells) [41,42,43,46,48,50,51,105]. Secondly, in budding yeast, the assembly of the precursor to the actomyosin ring is initiated very early in the cell cycle after the time of selection of the nascent bud site, but 4′-6′ prior to bud emergence (i.e., late G1) [35,41,42,43,44,50,52,105]. In mammalian cells the position at which the actomyosin ring will assemble is not determined until mitosis (M-phase) (specifically in metaphase/anaphase) [35,41,49,52]. In mammalian cells it is the position of the mitotic spindle that plays a key role in specifying the actomyosin ring position [35,41,49,52].

Thirdly, the actomyosin ring is essential for cytokinesis in mammalian cells but is dispensable for cytokinesis in budding yeast [35,41,42,50,52,105]. In mammalian cells the contractile force of the actomyosin ring drives the partitioning of the cytoplasm and cytokinesis, whereas the yeast actomyosin ring may primarily act to guide the deposition of the septum (septum deposition will be explained in more detail below) [35,42,43,45,46,47,48,49,50,52,105]. Fourthly, interestingly, contraction of the actomyosin ring is significantly slower in yeast than in mammalian cells [50]. This conclusion is based on the observation that despite their great difference in diameter the actomyosin rings of yeast and mammalian cells both contract completely within the same time frame (5–8 min) [46,48,50,105].

Another difference is that the coordination of mitotic exit with cytokinesis is regulated by the Mitotic Exit Network (MEN) signaling pathway in budding yeast. This pathway comprises Tem1p (a small GTPase); Lte1p (a GTP/GDP-Exchange Factor, GEF); Bub2p and Bfa1p, (a two-component GTPase-Activating Protein, GAP); Mob1p; Net1p; the protein kinases, Cdc15p, Dbf2p and Dbf20p; and the protein phosphatase Cdc14p [35,43,49,50,52,105]. A functionally equivalent pathway does not appear to exist in mammals although some components are present in mammalian cells (e.g., Cdc14) and, like their yeast counterparts, regulate cytokinesis [49,52]. Instead, in mammals physical interactions between the mitotic spindle and the contractile actomyosin cortex that underlies the plasma membrane (which is lacking in yeast) [50] play a major role in coordinating mitotic exit with cytokinesis [49,50]. Not only does budding yeast lack a contractile actomyosin cortex (except the actomyosin ring), but also (unlike in mammalian cells) the nuclear envelope remains intact during mitosis [35,49,52]. In budding yeast, the mitotic spindle is within the nucleus rather than in the cytoplasm like in mammalian cells and in yeast the nuclear envelope acts as a barrier between the mitotic spindle and the cell cortex [49,52]. There are other differences in regulatory mechanisms between mammals and yeast. For example, in mammals components of the chromosomal passenger complex (CPC), which plays roles in chromatin condensation, chromosome attachment to the mitotic spindle and mitosis (e.g., Aurora B kinase), are essential for initiation of cleavage furrow formation in cytokinesis [49]. In contrast, while the CPC exists in yeast components of this complex (e.g., the homologue of mammalian Aurora B kinase, Ipl1p) are dispensable for cytokinesis [49], although Ipl1p plays a role in the NoCut signalling pathway that inhibits cytokinesis if proper segregation of chromosomes does not take place during mitosis [49].

Moreover, it remains possible that some significant mechanistic differences in how the two rings contract underlie the especially rapid actomyosin ring contraction in mammals and the ability of actomyosin ring contraction to mediate cytokinesis in the absence of septum formation in mammals [50].

### 1.7. Initiation of Actomyosin Contractile Ring Assembly in Yeast

In late G1, following the formation of the septin ring at the nascent bud site the heavy chain of the type II conventional myosin (Myo1p) is recruited to the bud neck [35,41,42,43,50,52,105]. Myo1p recruitment and maintenance at the bud neck until the end of G2 is dependent on the septins [35,41,42,43,46,48,50]. The septin ring acts as a template or scaffold for assembly of a ring comprising Myo1p (but in G1 not yet F-actin or most other actomyosin ring components) [35,41,42,43,50,52]. This Myo1p ring lies within and immediately adjacent to the septin ring [35,43,48,50]. Upon S phase entry and bud emergence septins relocalize from the single septin ring to adopt a more diffuse hour-glass-shaped distribution (or “collar”) at the bud neck while Myo1p remains in a well-defined ring [35,42,43,48,50,87,89,90,91,97,106,107,108]. This Myo1p ring is not yet able to constrict because of the absence of F-actin (actomyosin ring constriction requires the presence of both F-actin and Myo1p) [35,41,42,43,50,51]. F-actin starts to localize in early anaphase to the Myo1p ring but localization is completed only in late anaphase/exit from mitosis to form a mature actomyosin contractile ring [35,41,42,43,44,49,50,52]. Upon exit from mitosis the septin hour glass (or collar) is remodeled (split) to form two septin rings such that one is placed on each side of the actomyosin contractile ring [35,43,48,50,87,89,91,97,106,107,108].

### 1.8. Role of the Formins Bni1p and Bnr1p in Actomyosin Ring Assembly in Yeast

During exit from mitosis the bud-neck-localization of the formin Bnr1p is gradually lost suggesting that either Bnr1p is degraded or it takes on a more diffuse distribution in the cell [35,50]. Meanwhile, the other formin, Bni1p, relocates from the cortex of the bud to the bud neck [35,50]. While both Bni1p and Bnr1p contribute to the assembly of a pool of linear actin filaments at the bud neck that is then assembled into the Myo1p ring to form a complete and contractile actomyosin ring, several lines of evidence suggest that Bni1p plays the predominant role [35,50,51,105]. The activity of Bni1p in actomyosin ring assembly is also under the control of the Rho-family GTPase Rho1p [35,49,50,51], which at this stage of the cell cycle is recruited to the bud neck and activated in a polo-like kinase (Cdc5p)-dependent process [35,50,109]. At this stage of the cell cycle (i.e., during mitotic exit) Rho1p is active, while the other Rho-family GTPase involved in bud formation, Cdc42p, is inactive [35,50,109,110,111]. Inactivation of Cdc42p is necessary for proper septum formation and cell separation [110] and prevents new bud emergence during cytokinesis, which could result in the death of the daughter cell [111]. In addition to Myo1p and F-actin, other proteins of the budding yeast actomyosin ring include the essential IQGAP-family protein Iqg1p/Cyk1p [35,41,43,44,50,51,52,110,112,113] and two myosin light chains (of which one, known as Mlc1p, is essential and is the light chain for Myo1p, Iqg1p/Cyk1p and Myo2p [41,50,52,114] and the other, known as Mlc2p, is non-essential and is the regulatory light chain for Myo1p) [35,50,52,114]. Iqg1p/Cyk1p [43,44] and its light chain Mlc1p [114] play a critical role in actomyosin ring assembly. Because the *IQG1*/*CYK1* and *MLC1* genes are essential (whereas the actomyosin ring is not) and *iqg1*/*cyk1* and *mlc1* mutants exhibit an arrest phenotype characteristic of a block in cytokinesis Iqg1p/Cyk1p and Mlc1p are likely to have additional functions in cytokinesis [44,52,113,114]. Both of the yeast tropomyosins (Tpm1p and Tpm2p) are also components of the actomyosin ring [56,109].

### 1.9. Contraction of the Actomyosin Ring

Once assembly of the actomyosin ring is completed after exit from mitosis (telophase), the ring undergoes contraction. Actomyosin ring contraction progresses gradually resulting in a narrowing of the diameter of the actomyosin ring until a small dot at the bud neck is all that remains [42,43,46,48,115]. The requirement for MEN-dependent exit from mitosis [108,116] for proper assembly of F-actin into the actomyosin ring [43] and the dependence of actomyosin ring contraction on F-actin [42] ensures that contraction does not commence until after exit from mitosis. In addition to the role of MEN in exit from mitosis, some individual MEN components also have roles in actomyosin ring contraction and completion of cytokinesis that become apparent when their upstream role in mitotic exit is experimentally bypassed, e.g., Tem1p (a Ras-family GTPase) [108], Mob1p [117] and Cdc15p (a protein kinase) [118,119].

Mlc2p is not required for actomyosin ring contraction, but appears to promote disassembly of the actomyosin ring during or following contraction [35,52,114]. Disassembly has been proposed to play a role in ring contraction and consistent with this some studies found that disassembly dependent on the Myo1p myosin motor domain and Mlc2p (although not essential) contributes to actomyosin ring contraction [50,115]. However, some other studies found no defect in actomyosin ring contraction in Mlc2p-deficient cells [114]. Coincident with or soon after contraction of the actomyosin ring, deposition of new cell membrane (referred to as membrane ingression) and cell wall material (referred to as septum formation) occur [42,43,45,46,47,48,52,89,106,120]. These involve localized membrane vesicle fusion events [45,48,120]. These membrane vesicles are transported by the type V myosin Myo2p along actin cables to the site of actomyosin ring contraction in a process similar to that used for transport of membrane vesicles to the growing bud [56,65]. Together, the processes of actomyosin ring contraction on the one hand and vesicle fusion, membrane ingression and septum formation on the other bring about the separation of the cytoplasm of the mother cell and the bud (cytokinesis) [42,43,44,45,46,47,48,89,105,106,120,121,122,123,124] (reviewed in [35,49,50,52,125,126]).

Before the actomyosin ring contracts during cytokinesis the septin hour-glass-shaped (collar) distribution is converted into a pair of clearly defined and well-separated septin rings that delimit the bud neck. One septin ring is positioned on the mother cell side of the bud neck and the other septin ring is positioned on the bud side of the bud neck [48,87,89,97,106] (reviewed in [35,50,91,107,108]). This conversion is thought to coincide with a switch in septin filament orientation from parallel to the mother cell-bud axis to perpendicular to the mother cell-bud axis [127,128] (reviewed in [35,91]). This switch is thought to be regulated by post-translational modifications of several individual septins [91]. However, not all the cytokinesis functions of septins require filament formation or filament arrays [88].

The molecular event that triggers commencement of actomyosin ring contraction is not yet known. It has been proposed that Tem1p could trigger actomyosin ring contraction [108]. In support of this possibility, Tem1p is known to physically interact with Iqg1p/Cyk1p by binding to the Iqg1p/Cyk1p GAP-related domain [113]. The Iqg1p/Cyk1p GAP-related domain is also required for actomyosin ring contraction, but not for actomyosin ring assembly [113]. As Tem1p is a GTP-binding protein, it is possible that binding to the Iqg1p/Cyk1p GAP-related domain triggers GTP hydrolysis on Tem1p leading to actomyosin ring contraction.

A second event that likely plays a role in triggering actomyosin ring contraction is the remodeling of the septin distribution at the bud neck. Initially, the septins have an “hour-glass-shaped” distribution (collar) that encompasses the actomyosin contractile ring on the outside and separates the actomyosin ring from the plasma membrane. This is remodeled to a double ring distribution in which each distinct septin ring lies on a different side of the bud neck with the actomyosin contractile ring in between [35,48,50,87,89,91,97,106,107,108]. This septin remodelling at the bud neck is regulated by Tem1p [108], independently of its role in mitotic exit (as part of the MEN).

A third event that is also likely to play a role in triggering actomyosin ring contraction is the re-polarization of the secretory membrane trafficking pathway from the growing bud to the bud neck upon exit from mitosis and delivery of new membrane material and the enzymes required for septum formation to the bud neck [35,45,46,47,48,50,52]. The polarized secretion of integral membrane proteins like the chitin synthase Chs2p to the bud neck and colocalization of Chs2p with Myo1p is required for stabilization of the actomyosin ring so that the actomyosin ring contracts effectively and symmetrically and does not break during contraction [35,45,46,47,48,50,52]. Inactivation of the mitotic cyclin-dependent kinase (CDK) Cdk1p/Cdc28p, which triggers exit from mitosis, results in dephosphorylation of Chs2p and this enables its polarized secretion to the bud neck [35,50,119,122,126].

### 1.10. Septum Formation during Cytokinesis in Yeast

Coincident with or soon after the contraction of the actomyosin contractile ring is the deposition of cell wall material at the bud neck to facilitate division of the cytoplasm (cytokinesis) [35,42,43,44,45,46,47,48,49,50,52,89,105,106,120,121,125,126]. Budding yeast cells, unlike mammalian cells, are surrounded by a rigid cell wall composed of polysaccharides [125,126]. The cell wall is 110–200 nm thick and comprises an outer and an inner layer [126]. There are three main classes of polysaccharide in the budding yeast cell wall: β-glucan (a branched polymer of glucose comprising both long β1,3-linked glucose chains and long β1,6-linked glucose chains cross-linked by glycosidic bonds) (30–60%) [126], chitin (a linear polymer comprising β1,4-linked *N*-acetylglucosamine residues) (1–2%) and mannan (a mannose-rich polysaccharide with mostly α1,6- but with some α1,2- and α1,3-linked mannose residues) [125,126]. Some β-glucan chains are linked by glycosidic bonds to the ends of some chitin chains [125,126]. Mannans are covalently attached to secreted and cell surface (including cell wall) proteins to form mannoproteins [126]. As well as mannoproteins, the cell wall also contains proteins that are covalently bonded to β1,3-glucan chains or β1,6-glucan chains, either directly, through amino acid residue side chains, or indirectly, through the degradation product of a glycosylphosphatidylinositol (GPI) anchor that had been attached to the protein post-translationally [126]. The outer cell wall layer comprises mannoprotein and its surface appears “brush-like”, while the inner cell wall layer comprises β-glucan and appears “microfibrillar” [126]. The entire yeast cell is enclosed by a thick cell wall, however the term septum refers specifically to the cell wall material deposited between the mother cell and bud to complete cytokinesis [125,126].

A ring of chitin is deposited at the nascent bud site in late G1 and remains at the neck as the bud emerges and this chitin is synthesized by chitin synthase III (the catalytic subunit is Chs3p) [52,87,120,124,125,126]. Septum formation between the mother cell and bud initiates with the deposition of a primary septum [125,126]. Localized deposition of the primary septum occurs from the outer boundary of the wide bud neck gradually inwards towards the centre of the narrowing bud neck (i.e., centripetally) and follows close behind the contracting actomyosin ring [35,42,43,44,45,46,47,48,49,50,52,89,105,106,120,121,124,126]. The conclusion of primary septum formation leaves a small gap (~40 nm wide) that still connects the cytoplasm of the mother cell and bud [35,129]. The primary septum comprises largely chitin (with some protein) which is synthesized in situ predominantly by the cell surface enzyme chitin synthase II (Chs2p) [35,45,46,47,48,50,52,89,106,120,121,122,123,124,125,126]. Chs2p is delivered to the bud neck via the fusion of membrane vesicles that contain newly-synthesized Chs2p [35,45,48,50,52,119,120,121,122,126]. During mitosis Chs2p is phosphorylated by the mitotic Cdk (Cdk1p/Cdc28p) and this prevents its transport from the endoplasmic reticulum [35,50,119,126,130]. The transport of Chs2p from the ER to the Golgi and into Golgi-derived late secretory vesicles destined for the bud neck is triggered by its dephosphorylation. Dephosphorylation of Chs2p is mediated by the MEN protein phosphatase Cdc14p whose activity is in turn activated by the two MEN protein kinases Cdc15p and Dbf2p [35,50,121,122,126,130,131]. The bud neck localization of Chs2p is dependent on the septins (which by this stage of the cell cycle have formed two well-separated rings) and the actomyosin ring [35,46,48,52,89,106]. As well as its role in regulating Chs2p traffic to the bud neck, the MEN also regulates Chs2p association with the actomyosin ring and/or catalytic activity after delivery to the bud neck via phosphorylation by Dbf2p [132]. In *chs2* mutant cells lacking Chs2p function or temperature-sensitive septin mutants (*cdc12*) unable to localize Chs2p to the bud neck there is a defect in actomyosin ring stability and/or the kinetics of ring contraction [35,45,47,48,50,52]. This suggests that the deposition of the primary septum may play a role in stabilizing the actomyosin contractile ring and/or promoting its efficient contraction. The *chs2* mutant cells show a complete loss of primary septum formation and septation is achieved by deposition of the secondary septum (see below), i.e., remedial septation [47,125,126,133]. Remedial septation is dependent on the activity of chitin synthase III (Chs3p), which functions in bud formation, but relocalizes to the bud neck prior to septation [47,52,120,124,126].

While mammalian cells are not enclosed in a polysaccharide cell wall, they nevertheless adhere to an extracellular matrix that contains polysaccharides (albeit of different chemical composition compared to the yeast cell wall) and secreted proteins. There is evidence that, like the cell wall in yeast, the extracellular matrix plays a key role in cytokinesis in mammalian cells [134,135]. Therefore, the yeast primary septum can be thought of as analogous to the mammalian cell extracellular matrix [35].

The next phase of septum formation is the deposition of a secondary septum on both the mother cell and bud side of the primary septum [35,46,47,50,124,125,126,129]. This process is accompanied by inactivation of the Cdc42p-dependent cell polarization pathway that initiated bud formation and only occurs after actomyosin ring contraction and once the primary septum nears completion [35,50,110,111,129]. The secondary septa serve to thicken and strengthen the cell wall at the bud neck [125,126]. Importantly, they also finally close the gap that remains after primary septum formation is complete [35,129]. Thus, formation of the secondary septa completes the partitioning of the cytoplasm between mother cell and bud. The secondary septa contain predominantly β1,3-glucan [35,50,125,126,129]. β1,3-glucan of the septum is synthesized by two distinct β1,3-glucan synthases which have as catalytic subunits Fks1p (Gsc1p) and Fks2p (Gsc2p), respectively [35,50,126,129,136,137]. The secondary septa also contain chitin [124], β1,6-glucan and protein-linked mannan [35,50,125,126]. The chitin of the secondary septa is synthesized by chitin synthase III (with catalytic subunit Chs3p), although *chs3* mutant cells lacking chitin synthase III function are viable and still capable of forming secondary septa and completing cytokinesis [35,50,124,125,126]. Chitin synthase II (Chs2p) activity is important for (indeed, it was reported to be essential for [35,124,125,126], although it is not strictly essential for [126]) the viability and ability to complete cytokinesis of *chs3* mutant cells that lack chitin synthase III activity. The Rho-family GTPase Rho1p plays a major role in secondary septum formation [35,50,126,129,138]. GTP-bound Rho1p binds to and activates the β1,3-glucan synthases with catalytic subunits Fks1p and Fks2p involved in secondary septum formation (reminiscent of Rho1p-dependent activation of the formin Bni1p in actomyosin ring formation) [35,50,77,126,138,139,140,141]. Rho1p is also required for the trafficking of Chs3p to the bud neck [138], possibly due to its role in stimulating formin-dependent actin cable assembly.

### 1.11. Septum Degradation and Cell Separation in Yeast

Formation of separate mother and daughter cells after cytokinesis (a process known as cell separation) is dependent on degradation of the primary septum and some of the secondary septa material by the secreted chitin-degrading endochitinase Cts1p and the secreted β-glucan-degrading endo-β1,3-glucanases Dse4p (also known as Eng1p) and Scw11p, which localize to the side of the septum closest to the bud and/or show bud-specific gene expression [125,126,142,143,144].

In contrast to humans (in which cytokinesis is dependent on the contractile force of the actomyosin ring), yeast has two independent pathways by which cytokinesis can be achieved: one is dependent on the actomyosin contractile ring and the other pathway is dependent on septum deposition and requires the product of a gene known as Hof1p. Hof1p is so-named because it is the budding yeast homolog of a *Schizosaccharomyces pombe* cytokinesis protein which is the product of the gene *CDC15 (Cell Division Cycle 15)* [42,85,105,145,146].

## 2. Budding Yeast Hof1p and Human PSTPIP1

### 2.1. The Function of Budding Yeast Hof1p and Fission Yeast Cdc15 and Interactions of the Hof1p SH3 Domain

We will now consider the function of Hof1p in cytokinesis and actin polymerization and how using yeast as a model organism could help to reveal the function of PSTPIP1, a mammalian homologue of Hof1p (Table 1). Cdc15 is a key regulator of cytokinesis in fission yeast (*S. pombe*) and has been shown to localize to the cleavage furrow of cytokinetic cells and to inhibit cytokinesis in both *S. pombe* and cultured mammalian cells when highly overexpressed [145,147]. The *S. pombe* gene *CDC15* that encodes Cdc15 is an essential gene (so deletion of the *CDC15* gene is lethal). Therefore, one has to use a conditional mutant to characterize the phenotype conferred by loss of Cdc15 function in *S. pombe*. *cdc15-140* is a temperature-conditional mutant which has no (or little) phenotype at room temperature but has a mutant defect after shift to restrictive temperature (36 °C). This is because *cdc15-140* cells express a mutated form of the Cdc15 protein that is functional at room temperature but non-functional at the restrictive temperature. Upon shift to the restrictive temperature *cdc15-140* cells become defective in actin ring formation and F-actin undergoes a change in subcellular localization. At restrictive temperature F-actin is dispersed throughout the mitotic cell and does not accumulate at the septum. This suggests a role for Cdc15 in actin polarization to the medial division site during cytokinesis [145,148].

Cdc15 protein expression is cell-cycle dependent and reaches a maximum level during cytokinesis. The *S. pombe* Cdc15 907 aa protein features a coiled-coil domain at the N-terminus, a motif rich in proline (P), glutamic acid (E), serine (S) and threonine (T) (PEST motif) that signals proteolytic destruction and a C-terminal Src Homology 3 (SH3) domain [145,149]. The presence of N-terminal F-BAR and coiled-coil domains is a characteristic of the family of F-BAR proteins. The term F-BAR domain was introduced when the amino acid sequence homology between N-BAR domains and a protein domain that consisted of an N-terminal FCH (Fes/CIP4 homology) and a coiled-coil (CC) domain was recognized [150]. The alternative term “extended FC (EFC) domain” stressing the connection between the FCH and the CC region is less used than “F-BAR domain”, because the term “BAR domain” has become strongly associated with membrane modeling [151].

F-BAR proteins combine membrane curvature with actin-assembly-driven processes allowing for cytokinesis, cell motility, endocytosis and exocytosis [152,153,154,155]. The original BAR domain (BIN1/Amphiphysin/Rvs167p) is found in the yeast proteins Rvs161p and Rvs167p and the mammalian proteins BIN1 (Table 1) and amphiphysin and has a role in direct membrane lipid binding and, dependent on the presence of other domains, either the introduction of curvature into membranes or the sensing of curvature in membranes. SH3 domains were first discovered as a conserved sequence in the oncogene Src [156] and mediate specific binding to short proline-rich core motifs such as “PXXP” [157,158] allowing for the regulation of dynamic processes such as signaling through the eukaryotic actin cytoskeleton and actin polymerization [159,160]. The PEST motif allows for rapid Cdc15 degradation after cytokinesis [149].

In budding yeast (*S. cerevisiae*) a deficiency in Hof1p due to deletion of the *HOF1 (Homolog of Fifteen 1)* gene (*hof1Δ*) results in slow growth and inefficient cytokinesis at the permissive growth temperature and cytokinesis failure at the restrictive temperature (a stress condition). After shift to the restrictive temperature the cells are unable to divide due to the lack of septum deposition at the bud neck. This results in the formation of chains of multinucleated cells with one continuous cytoplasm [85,105,146,161]. Moreover, F-actin straining (using rhodamine-phalloidin) revealed varying degrees of actin polarization defects in *hof1Δ* cells at the restrictive temperature, ranging from normal actin patch polarization to partial polarization defects [85,105,146]. In one study cortical actin patches accumulated at the bud tip in *hof1Δ* cells as in wild type cells, but during cytokinesis they did not properly repolarize to the mother bud-neck as in wild type cells [85]. Another study found a lack of repolarization of cortical actin patches to the bud neck in *cdc15-1 hof1*Δ *GAL1-SIC1^S^* cells (although repolarization appeared normal in *hof1*Δ *GAL1-SIC1^S^* cells [Note: *GAL1-SIC1^S^* promotes exit from mitosis]) prior to cytokinesis but did not find a defect in the assembly of the F-actin ring [131]. In contrast, two other studies that used formaldehyde-fixed *hof1Δ* cells found that the cytoskeleton appeared normal at both permissive and restrictive temperatures with properly polarized actin patches and a contractile actin ring [105,146]. Nevertheless, all these studies confirmed a defect in cytokinesis in these cells.

A quantitative comparison of the fluorescence intensities of F-actin staining in the cortical actin patches of fixed *hof1Δ* and wild type cells showed that Hof1p is not required for F-actin assembly in cortical actin patches [162]. This conclusion was supported by the in vivo analysis of endocytosis (a process driven by F-actin assembly at cortical actin patches) using live cell imaging of cortical recruitment, endocytic function and loss from the cortex of Sla1p-GFP and Abp1p-mRFP, which are markers of early- and late-phase steps of endocytosis at cortical actin patches, respectively. Also, staining of F-actin in fixed *hof1Δ S. cerevisiae* cells revealed a normal polarized distribution of cortical actin patches. However, in these fixed *hof1* mutant cells a defect in the arrangement of cytoplasmic actin cables was apparent with many cells showing disordered actin cables (i.e., not polarized along the mother cell-bud axis as in wild type cells). Analysis of actin cables in live *hof1Δ* mutant cells expressing the actin cable marker protein Abp140-GFP not only confirmed the defect in actin cable arrangement, but furthermore provided evidence that the actin cables grow transversely in *hof1* mutant cells rather than along the mother cell-bud axis as in wild type cells. This study also revealed an aberrant (discontinuous) staining pattern of Abp140p-GFP along the length of the actin cables suggesting an altered actin cable ultrastructure [162].

It was perhaps not unexpected that Hof1p plays a role in actin cable assembly and morphology as the linear actin filaments in actin cables are assembled by formins and Hof1p was originally discovered through the physical interaction of its SH3 domain with the proline-rich formin-homology 1 (FH1) domain of the formin known as Bnr1p [85]. The FH1 domains of the formins Bni1p and Bnr1p bind to profilin, an actin-monomer binding protein, and function in cell polarity and assembly of linear actin filaments [69,70]. It was found that Hof1p dimerizes through its F-BAR domain and its SH3 domain binds to the FH1 domain of Bnr1p, thereby inhibiting actin polymerization. The loss of this inhibition alters actin cable arrangement and functionality [162].

In wild type cells, both the expression and subcellular localization of Hof1p are subject to cell cycle regulation. Hof1p expression first becomes apparent in G2/M phase. Initially, Hof1p localizes to a pair of well-separated rings at the bud neck such that one ring lies on the mother cell side of the bud neck and the other on the daughter cell side of the bud neck. This subcellular distribution resembles that of the septins, which during G2/M also localize to a pair of well-separated rings at the bud neck. Indeed, the Hof1p rings are coincident with the septin rings and Hof1p localization to these rings is dependent on septin function [105].

During cytokinesis (i.e., telophase or exit from mitosis), Hof1p is phosphorylated by Cdc5p and then by the conserved MEN-kinase complex Dbf2p-Mob1p. This phosphorylation is essential for the release of Hof1p from the two septin rings and its relocalization to one ring that is placed precisely at the bud neck. This Hof1p single ring is adjacent to the actomyosin ring. However, while the actomyosin contractile ring contracts down to a dot the Hof1p ring only contracts slightly before gradually growing more diffuse. This suggests that the single medial Hof1p ring and the actomyosin ring are adjacent, but that the two rings are distinct [105]. There is increasing evidence that SH3-dependent physical interactions of Hof1p are important for actomyosin ring constriction. Hof1p has an important role in promoting actomyosin ring contraction, septum formation and membrane ingression [163,164]. Interestingly, this promotion of actomyosin ring contraction is believed to be achieved through binding of the Hof1p SH3 domain to proline-rich motifs while the unbound Hof1p SH3 domain may inhibit actomyosin ring contraction [164]. While the SH3 domain interactions responsible for inhibition of actomyosin ring contraction were not investigated in this study, a previous study found that the Hof1p SH3 domain binds three proline-rich motifs in Vrp1p. Vrp1p is a protein that together with the WASP homologue Las17p (and Myo5p) assists in nucleation of the assembly of branched actin filaments. It has been suggested that one role of Vrp1p in promoting the assembly of branched actin filaments involves its interaction with the Hof1p SH3 domain to counteract the inhibitory effect of the unbound Hof1p SH3 domain [165].

Taken together, this data suggests that the inhibitory effects of the Hof1p SH3 domain on actomyosin ring constriction and the nucleation of both linear (via Bnr1p) and branched (via Vrp1p) actin filaments strongly contribute to the regulation of actin polymerization and cytokinesis in yeast cells. The binding of the Hof1p SH3 domain to proline-rich motifs might also be implicated in the regulation of septum formation. The SH3 domain of Hof1p binds to proline-rich motifs in Inn1p, a protein essential for primary septum formation and regulation of the chitin synthase Chs2p, and Cyk3p, a protein that couples septum formation with membrane ingression [166,167](Figure 2).

### 2.2. The Function of Mammalian PSTPIP1 and Interactions of the PSTPIP1 SH3 Domain

Bearing a domain structure similar to Hof1p, the mammalian Hof1p homologue, PSTPIP1, comprises an N-terminal F-BAR domain (with a Fes/Cip4 Homology or FCH domain and coiled-coil domains), a PEST sequence and a C-terminal SH3 domain [168]. PSTPIP1 was discovered in a yeast two-hybrid screen as an interaction partner of the PEST protein tyrosine phosphatase (PTP-PEST) [147].

PSTPIP1 is predominantly expressed in hematopoietic cells. Inherited mutations in the gene that encodes PSTPIP1 are known to result in human disease. Two missense mutations with autosomal dominant inheritance that affect the coiled-coil domain of PSTPIP1 cause a disorder characterized by destructive inflammation of the skin and joints known as pyogenic arthritis, pyoderma gangrenosum and acne (PAPA syndrome) [155]. PSTPIP1 colocalizes with F-actin in the cleavage furrow, which is the mammalian equivalent to the bud neck in yeast, in dividing cells during cytokinesis and with the cortical actin cytoskeleton, e.g., lamellipodia, in non-dividing cells. Overexpression of PSTPIP1 in mouse fibroblasts causes extended filopodia, indicating a role of PSTPIP1 in actin polymerization [147]. Moreover, PTP-PEST-deficient fibroblasts exhibit hyperphosphorylation of PSTPIP1 and a cytokinetic defect, underlining the importance of PSTPIP1 for mammalian cytokinesis and a possible regulatory role for PSTPIP1 phosphorylation in cytokinesis [169,170].

Mammalian PSTPIP1 inhibits cytokinesis when overexpressed in *S. pombe.* Consistent with a cell division cycle defect, many of the *S. pombe* cells become elongated. The *S. pombe* cells form septa after mitosis but the septa are not cleaved so the daughter cells never separate from the mother cells. This results in the formation of a chain of attached cells with uncleaved septa and one nucleus in each cell. PSTPIP1 accumulates in the cleavage furrow of post-mitotic cells and at the ends of post-cleavage cells where it co-localizes with cortical F-actin [147]. However, despite the similar subcellular localization and function in cytokinesis of mammalian PSTPIP1 and *S. pombe* Cdc15, expression of mammalian PSTPIP1 in *S. pombe* does not rescue the cytokinesis defects of *cdc15Δ* cells [145,147]. Moreover, overexpression of PSTPIP1 in wild-type *S. pombe* cells results in a dominant-negative inhibition of cytokinesis completion [147].

The PSTPIP1 SH3 domain binds to Wiskott-Aldrich Syndrome Protein (WASP) [171], a proline-rich protein specific to hematopoietic cells that binds monomeric actin and interacts with WASP-interacting protein (WIP) [172]. Mutations in WASP that impair the physical interaction with WIP are the cause of Wiskott-Aldrich Syndrome, an inherited immunodeficiency disorder [173]. WASP is known to bind to the Arp2/3 complex and promote its activity in nucleation of actin filament assembly [68,174,175,176,177]. Some WASP-family proteins have also been shown to bind to actin filaments (F-actin) through a domain distinct from the domain that binds monomeric actin (G-actin) and has in vivo actin filament bundling activity [175,178,179,180].

Binding of the PSTPIP1 SH3 domain to proline-rich motifs in WASP has been shown to negatively regulate the F-actin bundling activity of WASP [171]. To investigate how this binding is regulated, a conserved tyrosine (Tyr367) in the SH3 domain of PSTPIP1 was substituted with aspartate or glutamate to mimic a negatively charged phosphate group. This substitution abolished interaction of the SH3 domain with WASP as assessed by in vitro binding assays. Abrogation of the PSTPIP1 SH3 domain binding to WASP in turn abolished colocalization of FLAG-tagged PSTPIP1 and GFP-tagged WASP in co-transfected CHO (Chinese Hamster Ovary) cells [171]. However, a subsequent study that employed co-immunoprecipitation of PSTPIP1 from cell lysates followed by phosphopeptide mapping found that the PSTPIP1 SH3 domain is not phosphorylated on Tyr367 [170]. It was concluded that PSTPIP1 serves as a scaffold, bringing WASP and PTP-PEST together and allowing the dephosphorylation of WASP by PTP-PEST. However, despite the lack of agreement about which tyrosine residues are phosphorylated these two studies do agree that there might be multiple tyrosine residues in PSTPIP1 that are phosphorylated and that phosphorylation of some of these tyrosines in PSTPIP1 is dependent on prior phosphorylation of PSTPIP1 on Tyr344, which is the major phosphorylation site in PSTPIP1 [170,171]. In contrast to Hof1p in yeast, which is phosphorylated during telophase, for PSTPIP1 in mammalian cells the timing of phosphorylation during the cell cycle is not known.

Additional studies using mice, a human macrophage cell line and fibroblast-like monkey kidney COS cells support the idea that the PSTPIP1 SH3 domain binds to WASP and negatively regulates nucleation of actin filament assembly [171,181]. Deletion of the equivalent proline-rich region in murine WASP causes actin cytoskeletal defects and impaired cell polarity in haematopoietic cells. This results in immunological deficiency in mice due to an inability of T lymphocytes to form an immunological synapse. This finding shows that the proline-rich region is critical for the immune function of WASP [182]. In this context it is interesting to note that in humans, the immunodeficiency in WAS has also been attributed to actin cytoskeletal defects in T lymphocytes [183,184,185,186,187].

### 2.3. How the Yeast Model Can Provide Insight into the Function of PSTPIP1 in Mammals

Similar to the mechanism of WASP binding to WIP in mammals, the yeast homologue of human WASP (Las17p) binds the yeast homologue of human WIP (Vrp1p) to promote Arp2/3p-dependent nucleation of actin filament assembly [188,189]. Moreover, the Hof1p SH3 domain has been shown to bind to three tandem proline-rich motifs in Vrp1p. These tandem proline-rich motifs are referred to as the Hof One Trap (HOT) domain. Binding of the Hof1p SH3 domain to the Vrp1p HOT domain promotes actin-filament-assembly driven processes, e.g., cytokinesis [165,190].

It has been shown that expression of human WIP in yeast *vrp1-1* mutant cells suppresses the temperature-sensitive growth defect, suggesting that human WIP and yeast Vrp1p are functional homologues [191]. Consistent with this, domains of human WIP required for the suppression of the growth defect in *vrp1-1* mutant yeast cells have been identified and include the WH2 actin-binding domain as well as the conserved proline-rich motif APPPPP that resembles an Actin-Based Motility (ABM) domain [189,190].

Expression of human WASP is unable to suppress the growth defect displayed by *las17Δ* yeast cells. However, expression of human WASP and WIP in combination is able to rescue the *las17Δ* mutant growth defect. This finding supports the idea that human WASP is the functional homologue of yeast Las17p. It also suggests that WASP functions better in yeast with human WIP than with yeast Vrp1p [192]. A possible mechanism would be that proline-rich motifs in WIP and/or WASP might rescue the growth defects of these mutant yeast cells at least in part by binding the Hof1p SH3 domain and thereby counteracting its inhibitory effect on actin-polymerization-driven processes.

F-BAR proteins such as PSTPIP1 dysfunction underpins a range of diseases such as neurodegenerative disorders, cancer, autoimmune and auto-inflammatory disorders [153,154,155,193]. The yeast model is helping to unravel the molecular mechanisms that contribute to these diseases and will facilitate the development of new treatments, e.g., based on counteracting the inhibitory effect of SH3 domains on nucleation of actin filament assembly.

## 3. Communication between the Actin Cytoskeleton and Protein Synthesis Machinery

### 3.1. Links between Actin and Translation

The actin cytoskeleton appears to be a hub for relaying various signals from both internal and external sources to the activity of important enzymes and signalling pathways [32,194]. A significant proportion of mRNAs, ribosomes, aminoacyl-tRNA synthetases and some translation factors are anchored to the actin cytoskeleton. This suggests that the actin cytoskeleton acts as a scaffold for the translation machinery. In addition, this association could provide a means for the actin cytoskeleton to spatiotemporally control the rate of protein synthesis. Supporting this idea, perturbation of the actin cytoskeleton leads to a dramatic reduction in the rate of global protein synthesis in both yeast and mammalian cells [194].

Recently, new insights have been provided into the mechanism that underlies this regulation. One of the most important insights involves the protein kinase Gcn2 (also known as eIF2AK4 in humans) (Table 1), which regulates protein synthesis via phosphorylation of the alpha subunit of eukaryotic translation initiation factor 2 (eIF2α) [195]. This link between the actin cytoskeleton and Gcn2 was first found in *S. cerevisiae*, and only subsequently in mammalian cells [196]. The existence of links between the actin cytoskeleton and Gcn2 in both yeast and mammals underscores the high degree of conservation of this crosstalk within the eukaryotic kingdom [196,197]. The actin-Gcn2 crosstalk is mediated by at least two actin-binding proteins which act as negative regulators of Gcn2 [194,198,199]. One of these negative regulators in mammals is the product of the imprinted gene with ancient domain (IMPACT) and in *S. cerevisiae* is the product of the yeast IMPACT homologue (*YIH1*) gene (Yih1p) (Table 1) [197,200,201]. The second negative regulator is the essential eukaryotic translation elongation factor eEF1A (Table 1) which amongst many other functions, regulates Gcn2 and also affects actin cytoskeletal dynamics [197,199,202].

### 3.2. Gcn2 Function

A constant supply of amino acids supports continuous protein synthesis; when amino acid supplies become scarce mechanisms must be activated to ensure cell survival. Gcn2 and the signalling pathway governed by Gcn2 have a well-recognized role in enabling cells to cope with and adjust to amino acid starvation [195,196]. Under conditions of starvation for one or more amino acids, the corresponding tRNAs cannot be aminoacylated and therefore accumulate as deacylated tRNAs (tRNAs^deacyl^); these tRNAs^deacyl^ function as a direct signal for Gcn2 activation. Gcn2 has a domain with sequence homology to histidyl-tRNA synthetases (the HisRS-like domain) [195]. This HisRS-like domain is not enzymatically active, but specifically binds tRNAs^deacyl^ [203]. Interestingly, the current working model predicts that Gcn2 detects ribosome-bound rather than free tRNAs^deacyl^ for induction of the signaling event. How Gcn2 senses the starvation signal is not known yet. A number of studies support a hypothesis in which under starvation conditions, when a cognate amino-acylated tRNA is not available, the cognate tRNA^deacyl^ enters the aminoacyl acceptor site (A-site) of the ribosome to then be detected by Gcn2. This is analogous to the well-studied mechanism in prokaryotes where the protein RelA detects tRNA^deacyl^ in the A-site [204,205]. In eukaryotes, from yeast to humans, Gcn2 may have evolved to perform this RelA function [196].

Gcn2 activation requires direct physical contact with its effector protein Gcn1. This interaction is mediated by the N-terminal RWD domain of Gcn2 and the RWD binding domain (RWD-BD) in Gcn1 [206] (Figure 3). It is possible that the Gcn1-Gcn2 complex shuttles on and off the ribosomes to probe for tRNA^deacyl^ present at the A-site [196]. A number of studies suggest that Gcn1 is directly involved in the transfer of the starvation signal to Gcn2. Gcn1 could promote the binding of tRNA^deacyl^ to the A-site, deliver the tRNA^deacyl^ from the A-site to Gcn2, and/or serve as a scaffold protein to position Gcn2 close to the A-site to allow it to better detect tRNA^deacyl^ [206,207,208].

Gcn2 is kept inactive via auto-inhibitory intramolecular interactions. The binding of tRNA^deacyl^ results in conformational changes within Gcn2 that release these autoinhibitory interactions. This leads to the stimulation of the catalytic domain of Gcn2, which upon auto-phosphorylation phosphorylates its substrate, eIF2α [195,209,210,211,212,213,214]. Recently, it has been found that the ribosomal P-stalk is involved in mediating Gcn2 activation in response to amino acid starvation [215,216,217]. The link between uncharged tRNAs and the P-stalk remains to be determined in view of Gcn2 activation under amino acid starvation, in yeast and mammals. Nevertheless, studies do suggest that Gcn1 is essential for Gcn2 activation in yeast as well as mammals, implying that Gcn1-Gcn2 interaction is required as well [196].

eIF2 is essential for the initiation of protein synthesis since in its GTP-bound state it forms a trimeric complex with the initiator methionyl-tRNA to deliver this tRNA to the ribosome [218]. After each round of initiation of protein synthesis, eIF2 is released in its GDP-bound form and for the next round of translation; it must be recycled to its GTP-bound form. The recycling process is catalysed by the guanine nucleotide exchange factor eIF2B. Upon phosphorylation by Gcn2, eIF2 becomes a competitive inhibitor of eIF2B, dampening the rate of GDP-GTP exchange by eIF2B. This results in a reduced ratio of GTP-bound to GDP-bound eIF2 and thereby to reduced levels of the trimeric complex in the cell. As a consequence, the rate of global protein synthesis is reduced, thus lowering the general consumption of amino acids. Simultaneously, specific mRNAs are translated at increased efficiency. These mRNAs code for specific transcriptional regulators, such as Gcn4 in yeast or its mammalian counterpart protein called Activating Transcription Factor 4 (ATF4). This translational up-regulation is exerted by specialized upstream open reading frames (uORFs) found in the 5′ leader sequence of these specific mRNAs. These uORFs repress the translation of Gcn4/ATF4 when trimeric complexes are abundant; when trimeric complexes are scarce, the inhibitory function of the uORFs is overcome, resulting in increased efficiency of Gcn4/ATF4 translation. Gcn4/ATF4 then reprograms the gene expression profile to allow the cell to adjust to the adverse environmental conditions. For example, transcription of genes that code for key enzymes in amino acid biosynthetic pathways and amino acid transporters is induced [195,219].

Details of the molecular mechanisms that govern Gcn2 activation have been elucidated following extensive studies conducted on yeast as model organism [195,196,209,210,211,212,213,214]. Fewer studies have addressed Gcn2 activation in mammals but they suggest that the mechanisms are highly conserved throughout the eukaryotic kingdom. For example, mammalian Gcn1 is required for mammalian Gcn2 activation [197] and the amino acid residue in yeast Gcn1 (Arg2259) that is crucial for direct Gcn1-Gcn2 interaction is also found in mammalian Gcn1 [201,206,220]. Overexpression of a fragment of yeast Gcn1p that is sufficient for Gcn2 binding impairs Gcn1-Gcn2 interaction and Gcn2 activation in yeast [206]. Similarly, overexpression of the equivalent mammalian Gcn1 fragment impairs Gcn2 activation in mammalian cells [221].

Since starvation for any single (or more) amino acid(s) in yeast cells induces the biosynthesis of all the amino acids, the yeast Gcn2 signaling pathway was called general amino acid control (GAAC) [195]. In yeast, Gcn2 is the sole eIF2α kinase; however, in mammals Gcn2 is one of four eIF2α kinases activated by different stress conditions. The mammalian eIF2α kinases activate the so-called integrated stress response (ISR) since diverse stresses converge to induce eIF2α-dependent phosphorylation by eIF2α kinases [222].

### 3.3. Reciprocal Regulation of eEF1A and Actin, and the Link to Gcn2

The translation elongation factor eEF1A, in its GTP-bound form, delivers aminoacyl-tRNAs to the ribosome during the elongation phase of protein synthesis (Table 1) [218]. eEF1A has many non-canonical functions, one of which is the bundling of actin filaments [202]. Certain mutations in eEF1A, as well as overexpression of wild type eEF1A, lead to defects in the actin cytoskeleton [223]. This indicates that eEF1A is a critical player in the regulation of actin dynamics. On the other hand, it was shown that F-actin binding to eEF1A decreases the affinity of eEF1A for guanine nucleotide and it leads to an increase in the rate of GTP hydrolysis [224,225]. Ultimately this results in the loss of GTP bound eEF1A and thus to reduced translation. It appears that eEF1A is able to bind to either aminoacyl-tRNA or F-actin [226]. This implies that the roles of eEF1A in F-actin bundling and in protein synthesis are mutually exclusive and that actin can control protein synthesis by binding to eEF1A.

eEF1A has two pH-sensitive F-actin binding sites. An increase in pH weakens the interaction of eEF1A with F-actin while the affinity for aminoacyl-tRNAs increases. Thus, changes in pH may be a means for regulating eEF1A-F-actin association in response to specific cues [223]. pH mediated loss of F-actin binding by eEF1A may be either the cause or perhaps a consequence of specific disease states. For example, pH gradient reversal (i.e., intracellular alkalinization and extracellular acidification) appears to be a hallmark of cancer [227] and it is possible that pH gradient reversal is a key player in the growth and metastasis of tumor cells. It has been proposed that one mechanism for supporting cancer cell growth is the high intracellular pH which leads to the dissociation of eEF1A from F-actin. eEF1A can then bind to aminoacyl-tRNAs and enhance the rate of translation to support the fast growth of cancer cells [228].

Interestingly, eEF1A was found to bind directly to the C-terminus of Gcn2 [199]. eEF1A-Gcn2 interaction is lost in vivo under starvation conditions and in vitro by tRNA^deacyl^, suggesting a model in which eEF1A binds to Gcn2 and keeps it in its latent state when amino acids are plentiful. Under amino acid starvation conditions it dissociates from Gcn2 allowing for Gcn2 activation. Considering that F-actin binds eEF1A, this raises the intriguing possibility that F-actin by means of eEF1A modulates the threshold for Gcn2 activation in response to amino acid starvation or other cues. One could envision that in cancer cells, for example, eEF1A dissociation from F-actin due to high pH levels would inhibit Gcn2 and thereby ensures high rates of protein synthesis. This will be discussed further below.

### 3.4. Regulation of Gcn2 by Yih1p/IMPACT and the Link to Actin

Yih1p is a protein with an N-terminal RWD domain, as found in Gcn2, and a C-terminal domain called the ancient domain (Table 1 and Figure 3) [229]. The name “ancient domain” stems from the fact that while Yih1p itself is specific to eukaryotes, the ancient domain (whose function is still unknown) is also found in prokaryotes and is therefore ancient in terms of its evolution [229]. Overexpression of Yih1p, or overexpression of its RWD domain alone, strongly inhibits the capability of yeast to overcome amino acid starvation and grow [200]. It was shown that Yih1p prevents Gcn2p activation because Yih1p competes with Gcn2p for binding to Gcn1p [198]. Supporting the idea that the Gcn2 and Yih1p/IMPACT RWD domains utilize the same binding determinant in Gcn1, Gcn1 Arg-2259 is essential for direct binding to both Yih1p and Gcn2 [198,201,206].

Purification of protein complexes and genetic experiments suggest that Yih1p is in complex with G-actin and that when liberated from G-actin it binds Gcn1p and thereby inhibits Gcn2p activity [198,230]. It is not known yet whether Yih1p also binds to F-actin. Yih1p overexpression does not affect the actin cytoskeleton as judged by microscopy of cells stained with the F-actin-specific reagent phalloidin. This finding may suggest that Yih1p binding to actin does not modulate cycles of actin polymerization and depolymerization, but instead that the ratio of polymerized to depolymerized actin determines the amount of Yih1p associated with actin [198]. One cannot exclude the possibility that other actin-binding proteins compete with Yih1p for G-actin binding. In the absence of Yih1p (deletion of the *YIH1* gene), however, there is a higher proportion of cells not stainable by phalloidin when compared to cells containing Yih1p [198]. This may suggest a defect in the assembly of actin cables in a cell population lacking Yih1p. It is possible that this phenotype may arise by a delayed progression through the cell cycle, due to the lack of Yih1p, causing an accumulation of cells in a specific cell cycle stage in which actin cables are not abundant [231].

Yih1p is not a generic or constitutive Gcn2p inhibitor, but instead appears to down-regulate Gcn2p activity only under certain conditions and/or in specific locations in the cell (e.g., when or where it is not associated with G-actin) [198]. Yih1p was found to associate with ribosomes as found for Gcn1p and Gcn2p, suggesting that this allows instant Gcn2p inhibition at the site where Gcn2p senses the starvation signal and reversal of inhibition [232]. Therefore, one can envision that due to the changing dynamics of actin filament assembly and disassembly, the actin cytoskeleton spatiotemporally determines the amount of Yih1p that is available to dampen Gcn2p activation. This hypothesis arose from the finding that strains lacking Yih1p can grow on starvation as well as replete medium at the same rate as a wild-type strain. However, knock-down of actin (by deleting one of the two alleles in a yeast diploid strain) affects growth under starvation. Presumably this is because there is less G-actin available to sequester Yih1p, with the consequence that liberated Yih1p would bind to Gcn1p resulting in the inhibition of Gcn2p activation. Supporting this idea, the growth impairment under starvation conditions was rescued, at least in part, by deleting the *YIH1* gene in the actin knock-down strains [198].

Studies strongly suggest that mammalian IMPACT is the functional homologue of yeast Yih1p (Table 1), implying that the Gcn2 regulation by the Yih1/IMPACT family of proteins is evolutionary highly conserved. For example, IMPACT binds to G-actin in mammals as was demonstrated for Yih1p in yeast [197,198]. Heterologous overexpression of IMPACT in yeast, or IMPACT overexpression in mouse embryonic fibroblast cells (MEFs), impairs Gcn2 activation, and this is associated with reduced Gcn1-Gcn2 interaction, as found upon over-expression of Yih1p in yeast cells [198,201,221]. IMPACT associates with translating ribosomes in mammalian cells as found for Yih1p in yeast, which supports a model in which IMPACT-ribosome association brings IMPACT close to the ribosome-bound GCN1-GCN2 complex to efficiently impair GCN2 activation [232,233].

In mammals, the proposed spatiotemporal regulation of GCN2 appears to have an additional layer. This regulation involves the differential expression of IMPACT in certain organs and cell types during development. For example, IMPACT is highly abundant in neurons, and its expression increases upon neuronal differentiation [201,233,234]. IMPACT is highly abundant in the hypothalamus, which is critically involved in the maintenance of homeostasis, i.e., adjusting the organism’s metabolism to meet the organism’s immediate needs. Continuous protein synthesis may be paramount to sustain neuronal constant signaling processes in the hypothalamus. Thus, it is tempting to speculate that GCN2 activation - and the associated reduction in translation - must be prevented, even under conditions where GCN2 is activated in other cells [201].

Interestingly, the GCN2-IMPACT axis participates in neuronal differentiation [233]. The knock-down of IMPACT prevents neurite outgrowth, while GCN2-knockdown leads to spontaneous neurite outgrowth. This suggests that increased abundance of IMPACT is important for the development of neurons [233]. Interestingly, actin is involved in remodelling neurons as well, raising the possibility that the actin-IMPACT axis is critical for this process (see below).

### 3.5. Gcn2 is an Important Sensor of the State of the Actin Cytoskeleton

The links between actin, eEF1A, Gcn2p, Yih1p/IMPACT and Gcn1p that emerged from studies using yeast prompted a comprehensive study of the interdependence of these players in mammalian cells. Similar links to those discovered in yeast were revealed and in some cases more robust evidence has been obtained in mammals [197]. For example, it was found that in response to actin depolymerization GCN2 becomes more active, concomitantly leading to a reduction in translation. This effect appears to be due to two mechanisms. In the first mechanism, actin depolymerization leads to increased levels of tRNA^deacyl^, the activating ligand for GCN2. The increased level of tRNA^deacyl^ could be a consequence of: a) Impairing the function of amino acid transporters [235] and/or b) F-actin disassembly affecting the function of aminoacyl-tRNA synthetases which are part of an actin cytoskeleton-associated multiprotein complex [236]. The second mechanism is the shift in abundance of protein complexes relevant for GCN2 activation (Figure 3). In view of recent findings (see above) [215,216,217], it would be interesting to investigate whether the interaction between Gcn2 and the ribosomal P-stalk is affected by actin depolymerization.

Under conditions favoring F-actin assembly, GCN2 is kept inactive because GCN1 is mainly in a complex with IMPACT. Furthermore, eEF1A binding to GCN2 keeps GCN2 in its latent state. However, under conditions favoring actin depolymerization, IMPACT is increasingly found in complexes with G-actin rather than with GCN1, thereby making GCN1 more available for interaction with GCN2. The result is an increase in GCN1-GCN2 complex formation, which is likely to reduce the threshold for GCN2 stimulation by tRNA^deacyl^. On the other hand, actin depolymerization could also have an opposite effect on GCN2 activation since eEF1A released from F-actin could then potentially inhibit GCN2. However, the simultaneous presence of higher levels of tRNA^deacyl^ would reduce the sensitivity of GCN2 to this eEF1A-dependent inhibition, perhaps by causing eEF1A to dissociate more easily from GCN2. Support for this possibility comes from the finding that actin depolymerization displaces eEF1A from GCN2 [197]. The fact that eEF1A is released from F-actin upon actin depolymerization should enhance its recruitment to translating ribosomes and thereby promote translation. However, in contrast, the release of eEF1A from GCN2 would be expected to promote easier activation of GCN2, which in turn would be expected to dampen translation. These two effects may appear contradictory. However, one could envision that these counteractive mechanisms may be in place to juggle the fine balance between the appropriate levels of global translation versus levels of translation of specific mRNAs, such as those encoding ATF4. Given that depolymerization of the actin cytoskeleton is subjected to spatiotemporal regulation, GCN2 activity could be modulated according to local needs within a cell. A prime example would be neuronal development and synaptic function, which require localized synthesis of proteins [194,233]. F-actin remodelling could then also modulate the activity of Gcn2 in a mechanism dependent on IMPACT [197].

### 3.6. The GCN2-Actin Regulatory Axis May Have a Wide-Reaching Relevance

Both the global integrity and dynamics of the actin cytoskeleton, as well as specific actin-binding proteins, play a critical role in the formation of long-term memory (LTM) [237,238]. GCN2 has been implicated in memory formation, a process known to rely on localized actin rearrangements and spatiotemporal regulation of translation [237,238,239], which could both be influenced by IMPACT and eEF1A. Memory formation involves changes in synaptic strength which are dependent on the dynamic actin cytoskeleton [237,238]. The strengthening and facilitation of synaptic connections, known as long term potentiation (LTP), is a key process for the storage of information [237,238,239]. Weak training protocols lead to the early phase of LTP (E-LTP, lasting 1–2 h), and short-term memory (STM). Strong training, via repeated stimulation/activities, stimulates mechanisms that stabilize synaptic changes and results in late-phase LTP (L-LTP, lasting several hours), and long-term memory (LTM). While STM and E-LTP involve the modification of pre-existing proteins, LTM and L-LTP require the expression of new genes regulated at the transcriptional and translational level. Hence, conversion of STM to LTM requires de novo protein synthesis. Expression of genes required for L-LTP and LTM is mediated by the cAMP responsive element binding protein (CREB). CREB is under the control of repressor protein ATF4, and ATF4 expression is regulated at the level of mRNA translation through alteration of the levels of eIF2α phosphorylation. Induction of L-LTP correlates with decreased eIF2α phosphorylation. Based on studies with knock-out mice, it was proposed that GCN2 in neurons provides a basal level of phosphorylated eIF2α that allows a rate of ATF4 translation sufficient for suppressing CREB activity. Stimulation leading to L-LTP would involve a decrease in GCN2 activity thereby reducing the level of eIF2α phosphorylation, and hence relieving inhibition of CREB. This would, in turn, allow CREB-dependent expression of synaptic plasticity-related genes leading to LTM formation. Training regimens that normally do not lead to LTM do so in GCN2-deficient mice. Curiously, however, training paradigms that normally lead to L-LTP, fail to do so in GCN2-deficient mice. Thus, it appears that GCN2 regulates the switch from E-LTP to L-LTP, and hence from STM to LTM, and that reduction in eIF2α phosphorylation reduces the threshold for L-LTP and LTM formation. This suggests that GCN2-dependent regulation of ATF4 translation is required for the appropriate generation of LTM. However, the exact molecular mechanisms that underlie the fine-tuning of GCN2 basal activity remain to be elucidated. Interestingly, the induction of LTP increases the formation of F-actin at the cost of G-actin levels [237]. It is tempting to speculate that actin contributes to the spatiotemporal regulation of GCN2 and protein synthesis. This regulation could be achieved by control of the release of eEF1A from F-actin (and the recruitment of eEF1A by the protein synthesis machinery and inhibition of GCN2 activity) (see above). In support of this possible mechanism, it has been found that de novo eEF1A synthesis is increased during L-LTP [240]. Remodelling of the actin cytoskeleton in the process of LTP could also potentially result in the release of IMPACT from G-actin and consequently in increased IMPACT-GCN1 complex formation, leading to the inhibition of GCN2 activity.

GCN2 responds to various environmental stresses in mammals, including glucose starvation, rapamycin treatment, oxidative stress, tubulin depolymerization [196]. These stresses seemingly result in increased tRNA^deacyl^ levels, the ligand for Gcn2 kinase activation. In metazoans Gcn2 has been found to be involved in more advanced functions, such as metabolism, insulin signaling, the immune response defense against viral infection, determination of life span, cell cycle progression and initiation of developmental programmes. Gcn1 and tRNA^deacyl^ may also be important players for these Gcn2-dependent functions, and therefore eEF1A, IMPACT and actin may act as Gcn2 regulators as well. For example, the GCN2-IMPACT module has already been shown to play a role in the part of the immune system that regulates cellular responses via an ancient strategy, which is by controlling nutrient supply [196]. Dysfunction of Gcn2 has been implicated in diseases and disorders such as cancer [196,241,242] underscoring the need to better understand Gcn2 function and regulation to better treat and prevent diseases/disorders.

Yih1/IMPACT appears to have functions in addition to inhibiting Gcn2. Deletion of *YIH1* results in a delayed cell cycle. Yih1p binds to the cyclin-dependent kinase Cdc28p in stages of the cell cycle where Cdc28p is active [231]. This suggests that Yih1p is involved in the regulation of cell cycle progression by a mechanism that is dependent on Cdc28p. Yih1p also binds to the mammalian counterpart of Cdc28p, CDK1, suggesting evolutionary conservation of this regulation. Interestingly, this Cdc28p-dependent Yih1p modulation of the cell cycle is independent of Gcn2p [231]. Given that a malfunctioning cell cycle may be the cause for cancer, further investigations aiming to achieve a better understanding of the involvement of actin in these GCN2-independent processes of Yih1p/IMPACT are warranted.

## 4. The Yeast and Human Amphiphysins and Their Link to Actin-Based Cellular Functions

### 4.1. The Yeast Rvs161p and Rvs167p Amphiphysins, Key Regulators of Actin-Dependent Endocytosis

The yeast *S. cerevisiae* amphiphysins Rvs161p and Rvs167p are two closely related proteins involved in actin cytoskeleton organization, sporulation and endocytosis encoded by the Reduced Viability upon Starvation 161 (*RVS161*) and *167* (*RVS167*) genes (Table 1, Figure 4A). These genes were first identified in a screen for mutants that exhibited *r*educed *v*iability upon *s*tarvation [243]. Mutations in *RVS161* and *RVS167* result in similar phenotypes associated with a loss of viability and aberrant cell morphology in minimal or salt-rich medium growth conditions, delocalized actin distribution and abnormal (random) budding in diploid cells [243]. Their link to actin cytoskeleton and vesicular trafficking was identified in 1995 by the laboratories of David Botstein and Howard Riezman [244,245]. By using a two-hybrid screen, Amberg et al. [244] could show that the Rvs167p protein interacts with actin through its SH3 domain. Munn et al. [245] prepared a temperature-sensitive yeast mutant collection and screened the 220 mutants individually for a defect in endocytic internalization of the α-factor pheromone. They isolated the *end6-1* mutant that was allelic to *RVS161*. Sequence comparisons revealed that the yeast Rvs161p and Rvs167p proteins belong to the amphiphysin family of proteins [246] (Figure 4).

In previous studies it was shown that amphiphysins possess a BAR domain (see above). There are multiple types of BAR domain. The original BAR domain is the N-BAR, other types of BAR domains are the F-BAR and I-BAR [247]. The N-BAR domain is characterized by an N-terminal amphipathic α-helix that allows binding to lipids [247]. Rvs161p and Rvs167p are the only N-BAR-domain proteins in *S. cerevisiae*. Rvs161p and Rvs167p interact with each other to form heterodimers through their BAR domains. The heterodimer has a characteristic banana-shaped structure, able to sense membrane curvature [248]. This membrane binding is essential for the final stages of endocytosis as it promotes the internalization of endocytic vesicles at the plasma membrane [249].

The overall domain structure of Rvs161p and Rvs167p is different: Rvs161p consists only of an N-BAR domain, whereas Rvs167p is composed of an N-terminal N-BAR domain followed by a region rich in glycine, proline, and alanine (GPA) and a C-terminal SH3 domain. The GPA region is not conserved among the members of the amphiphysin family and may play a role in Rvs regulation because it is phosphorylated in vivo [250]. Domain mapping has shown that the BAR domain is required for Rvs167p functions in salt resistance, bipolar budding and endocytosis, except for sporulation where the SH3 domain is required. The N-terminal helix of the Rvs167p/Rvs161p BAR domain is required for high-affinity binding to phosphoinositide-enriched membranes. The BAR domain affects the fluidity of the membrane in the presence of phosphatidylinositol-4,5-bisphosphate (PtdIns(4,5)P_2_) [251].

However, distinct roles for the individual Rvs proteins have also been reported. The BAR domains of Rvs161p and Rvs167p are not interchangeable, since these BAR domains cannot be functionally replaced with each other [252]. The localization patterns of Rvs161p and Rvs167p determined by immunofluorescence microscopy are similar but not identical to each other as Rvs167p is localized to cortical actin patches, whereas Rvs161p is reported to be mainly cytoplasmic with small dots distributed randomly within the cell cortex in non-budded cells and at the mother-bud neck during bud emergence and cytokinesis [253], however the Rvs161p-GFP (green fluorescent protein) fusion has also been localized to small cortical patches during G_1_ phase [254]. The *fus7* mutant was identified in a screen for cell fusion mating mutants and revealed to be allelic to *RVS161*, however the role of Rvs161p in cell fusion is different from its role in endocytosis, since the *end6-1/rvs161-1* endocytosis mutant shows no defect in cell fusion. Moreover, the Rvs167p mutant strains show no defect in cell fusion and this role is specific for Rvs161p in complex with the cell fusion regulator Fus2p [254].

Live-cell imaging studies in yeast have allowed deciphering of the dynamics and functions of actin-dependent effectors during endocytic internalization [66]. These studies in yeast cells revealed the succession of steps leading to the internalization (initiation, invagination, scission and vesicle release) and the different protein complexes required at these different steps [255]. The current model for actin-based endocytic internalization relies also on immuno-electron-microscopy studies on yeast cells showing the position of the different effectors along the endocytic invagination [256].

Dynamic actin structures are associated with the endocytic vesicles and favor their formation, their release from the plasma membrane and their transport into the cell cytoplasm. The Rvs167p and Rvs161p N-BAR domains inhibit the lateral diffusion of PtdIns (4,5)P_2_ and generate extremely stable lipid microdomains by assembling into very stable scaffolds on PtdIns(4,5)P_2_-enriched membranes [251]. The N-BAR domain of Rvs167p interacts with calmodulin and this interaction is required for endocytosis by regulating its membrane remodeling activity [257]. The SH3 domain of Rvs167p directly interacts with the yeast dynamin-related protein Vps1p. This interaction appears to be important as in the absence of Vps1p a reduced level of Rvs167p and Rvs161p are recruited to the sites of endocytosis at the plasma membrane [104]. The endocytic vesicles are released from the plasma membrane by a scission process. The scission process requires the action of heterooligomers comprising the amphiphysin-related proteins Rvs167p and Rvs161p (which sense the membrane curvature via their BAR domains) and the dynamin-related protein Vps1p. In conclusion, yeast studies have revealed the in vivo role of Rvs161p and Rvs167p that are required for the internalization step of endocytosis and associated to actin filament assembly.

### 4.2. The AMPH1 and BIN1 Human Amphiphysins and Their Link to Actin Cytoskeleton

In the human genome there are two genes coding for amphiphysins, *AMPH1* and *BIN1* (Bridging Integrator 1) or *AMPH2*, which are homologous to yeast Rvs167p in function and in protein organization with similar domains (Table 1 and Figure 4). *AMPH1* encodes amphiphysin 1 that is concentrated in the central nervous system and required for clathrin-mediated endocytosis of synaptic vesicles [258]. *BIN1* encodes the ubiquitously expressed BIN1/amphiphysin 2 protein associated with different diseases [259]. Here, we will focus on human BIN1/amphiphysin 2 since recent data show that it has a functional link to the actin cytoskeleton [260,261,262,263,264]. The *BIN1* gene is located on chromosome 2q14.3 and encodes 20 exons, some of which are alternatively spliced, leading to 10 different isoforms and a tumor isoform 11 also termed BIN1 + 12A (Figure 4B) [259,265]. The largest isoform of BIN1 is referred to as the canonical BIN1 isoform 1 and is enriched in the central nervous system. BIN1 isoform 8 represents the most studied muscle-specific isoform of BIN1 (Figure 4B) [266].

All BIN1 isoforms have three conserved domains. The N-BAR domain consists of three α-helices and assembles into a homodimer with a “banana-like” curved structure that stabilizes the curvature of the membrane, without the need for heterodimeric amphiphysin complexes [267]. However, heterodimeric complexes between BIN1 and Amph1 amphiphysin were also observed but only in neurons, where they play a role in clathrin-mediated endocytosis by enhancing association with dynamin [268]. The N-BAR domain of BIN1 is also involved in direct interaction with actin, thereby regulating actin dynamics by stabilizing actin filaments [264]. The second domain is the MBD (Myc-Binding Domain) domain, which confers a tumor suppressor function. Indeed, through this MBD region, BIN1 can physically and functionally interact with the transcription factor encoded by the Myc oncogene and inactivate it [259]. The third domain is the SH3 domain at the C-terminus. This domain interacts with the proline-rich motifs (proline-rich domain or PRD) of proteins such as the dynamin-family protein encoded by *DNM2* and the PRD of BIN1 itself to mediate autoinhibition. The autoinhibition mediated by this SH3 domain is the basis for the functional regulation of amphiphysin-family proteins. Indeed, intramolecular interactions between the BIN1 SH3 and PRD domains inhibit BIN1 function in membrane remodeling. On the other hand, intermolecular interaction between the BIN1 SH3 domain and the PRD domain of dynamin relieves this autoinhibition of BIN1 function in membrane remodeling [269,270]. Expression of the BIN1 SH3 domain in mice induces disorganization in myofibers due to the association of this SH3 domain with actin and myosin filaments, and with the pro-myogenic Cdk5 kinase [261]. In skeletal muscles, the SH3 domain of BIN1 also interacts with N-WASP, a key regulator of the actin cytoskeleton dynamics [262].

In *C. elegans* and in human fibroblasts, BIN1 interacts with Nesprin2, a protein connecting the nuclear envelope to the actin cytoskeleton, and this binding only occurs with full-length BIN1 and not with its SH3 or BAR domains [260]. Plasma membrane repair, a critical process for muscular functions, implies reorganization of the actin cytoskeleton and requires formation of two different protein complexes: the annexins complex forming a repair “cap” and the “shoulder” protein complex including among others BIN1 and dysferlin [263]. The BIN1 domains required for this actin-based membrane repair remain to be identified.

Some domains are only present in specific BIN1 isoforms (Figure 4B). The PI (phosphoinositide)-binding domain is a polybasic domain encoded by exon 11 and is only present in the muscle-specific isoform 8 (Figure 4B). The PI domain interacts with a particular type of lipid phosphoinositide [the phosphatidylinositol-4,5-bisphosphate (PtdIns(4,5)P_2_)], which is enriched at the plasma membrane and regulates endocytosis. Thus, this PI domain facilitates the binding of BIN1 to the muscle cell membrane and is important for the ability of BIN1 to generate tubular plasma membrane invaginations [271]. Indeed, BIN1 plays a critical role in the membrane remodeling processes in muscle cells via the recruitment and regulation of its partner protein dynamin 2 [272,273]. The CLAP (Clathrin-Associated Protein) domain mediates BIN1 interaction with clathrin, is encoded by exons 13 to 16, and is present only in the BIN1 isoforms found in the nervous system (Figure 4B). The CLAP domain is involved in the binding of BIN1 to clathrin-coated membranes. However, targeted disruption of the *BIN1* gene in mice does not alter endocytosis but leads instead to embryonic cardiomyopathy [274].

### 4.3. BIN1 Associated Diseases and Their Link with Actin-Based Functions

#### 4.3.1. Cancer

It has been reported that BIN1 is often missing or functionally inactivated in melanoma, breast and prostate cancers. BIN1 interacts with the Myc box region at the N-terminus of the Myc oncoprotein transcription factor and inhibits c-Myc–mediated transactivation and oncogenic transformation. This interaction is mediated through the BIN1 Myc-binding domain (MBD) and deletion of the MBD leads to a failure of the BIN1 protein to inhibit the oncogenic activity of Myc [259]. In contrast, inhibition of Myc-BIN1 interaction in cells through overexpression of the MBD promotes oncogenic transformation and reduces the ability of Myc to induce apoptosis in primary cells. Recent data highlight the crucial role of nuclear actin in regulation of transcription, cell cycle and DNA repair [275]. Interestingly, c-Myc activation in medulloblastoma cells resulted in increased metabolic activity, changes in cellular morphology and F-actin cytoskeleton remodeling associated to cofilin nuclear translocation [276]. Therefore, impaired BIN1 functions associated with c-Myc oncogenic transformation could be linked to nuclear translocation of cofilin, an actin depolymerizing factor, and result in actin cytoskeleton remodeling and tumor metastasis.

Interestingly, Myc-independent BIN1 activity in tumor suppression has also been reported. For example, a mutated form of BIN1 lacking the MBD (BIN1ΔMBD) does not bind or inhibit c-Myc. However, overexpression of BIN1ΔMBD in primary fibroblasts obtained from rodents attenuates oncogenic transformation when Ras is co-transformed with genes encoding non-Myc nuclear oncoproteins, such as adenovirus E1A or dominant-negative mutant p53. The ability to attenuate oncogenic transformation by either of these non-Myc oncoproteins requires the BIN1ΔMBD SH3 domain [277].

The BIN1 SH3 domain also has an additional role in Myc binding, suggesting that BIN1 interacts directly with and suppresses the oncogenic activity of c-Myc via the SH3 domain as well as the MBD [278]. BIN1 activates a cell death program that is independent of caspase since a broad-spectrum caspase inhibitor did not inhibit this cell death [279]. Moreover, a part of the BIN1 BAR domain is also required for suppression of cancer growth, independently of c-Myc inhibition [280]. Thus, BIN1 exerts also its tumor suppression activity by Myc-independent mechanisms that could be linked to cytoplasmic and/or nuclear actin cytoskeleton regulation via its SH3 domain, as reported for BIN1 muscular functions.

#### 4.3.2. Centronuclear Myopathies

Mutations in the *BIN1* gene are responsible for the CNM2 type of centronuclear myopathy (CNM) [269]. CNM is an umbrella term used for a group of rare genetic muscle diseases associated with mutations in different genes (*MTM1*, *DNM2*, *BIN1* and *RYR1*) characterized by the presence of an abnormally high number of muscle fibers with central nuclei. The autosomal recessive form of CNM with the onset of weakness in infancy or early childhood with or without ophthalmoplegia (OMIM 255200) was termed CNM2 and is due to mutations in the *BIN1* gene. The analysis of the position of the mutations in the *BIN1* gene of patients with CNM2 showed that the SH3 and BAR domains are the main areas affected by point mutations [281].

In skeletal muscle, BIN1 is concentrated around transverse (T) tubules that function in the excitation-contraction coupling machinery of skeletal muscle cells [266]. BIN1 is involved in the induction of the membrane curvature leading to the formation of T-tubules [282]. Missense mutations in the BAR domain were shown to affect its membrane tubulation properties and this could alter the formation of the T-tubules [269]. Indeed, BIN1 clusters the lipid PtdIns(4,5)P_2_ in specific membrane sites to control the recruitment and accumulation of its partner protein dynamin [272].

A mutation found in a CNM patient that causes a partial truncation of the BIN1 SH3 domain abolishes the interaction of BIN1 with the dynamin Dnm2. Mutations in the *DNM2* gene are associated with an autosomal dominant form of CNM that usually appears in adulthood and is slowly progressive (OMIM 160150, CNM1) [283]. BIN1 acts as a negative regulator of DNM2 activity during muscle maturation and modulation of DNM2 intracellular levels alleviates the requirement for BIN1 since *Bin1^−/−^ Dnm2^+/−^* mice were alive and did not suffer from myopathy whereas the *Bin1^−/−^* KO mice were not viable [273]. Moreover, human BIN1 overexpression rescued the myopathy phenotypes displayed by the *Mtm1^−/y^* mice and BIN1 function in skeletal muscles is linked to focal adhesions by controlling integrin and laminin localization [284].

Mutations Q573X and K575X found in the SH3 domain of BIN1 in CNM2 patients (Q573X and K575X) abolish the interaction with the actin-cytoskeleton regulator N-WASP, and this in turn leads to mislocalization of the nuclei in the myofibers. Moreover, the localization of N-WASP is also altered in the muscle of a CNM2 patient with a BIN1 mutation outside the SH3 domain (R154Q), showing that mis-regulation of N-WASP by BIN1 is key in centronuclear myopathy pathophysiology [262].

Mis-positioning of the nuclei is a hallmark of CNM2 and in *C. elegans* downregulation of amphiphysin induces a mis-localization of nuclei. Moreover, interaction between human BIN1 and Nesprin2, a component of LINC (Linker of Nucleoskeleton and Cytoskeleton) complex is required for actin-dependent movement of nuclei and the CNM2 BIN1-K575X mutation affecting the SH3 domain alters nuclear movement. These data show that BIN1 through its SH3 domain is a key player in the regulation of nuclei position by linking the nuclear membrane to the actin cytoskeleton [260].

#### 4.3.3. Alzheimer’s Disease

Alzheimer’s Disease (AD) is pathologically defined by extensive neuronal loss and the accumulation of intracellular neurofibrillary tangles and extracellular amyloid plaques in the brain. Recent genome-wide association studies (GWAS) have identified *BIN1* as a susceptibility gene associated with AD [285], *BIN1* being one of the top candidate genes for susceptibility to late-onset AD (LOAD) [286]. In the brain, the largest BIN1 isoform (isoform 1, Figure 4B) is enriched in the central nervous system and is localized in the cytomatrix beneath the plasma membrane of axon initial segments and nodes of Ranvier [266]. In human AD diseased brains, the level of the transcript that encodes the largest isoform of BIN1 (isoform 1) is significantly reduced whereas the levels of the transcripts that encode the smaller BIN1 isoforms are increased [287].

In human neuroblastoma cells and in mouse brain, BIN1 interacts and colocalizes with the Tau (tubule associated-unit) protein that is associated with AD [288]. The sites of interaction between Tau and BIN1 were mapped to the SH3 domain of BIN1 and the PRD domain of Tau, with phosphorylation in and around the PRD decreasing the binding to BIN1 SH3 in vitro and in vivo [289]. Moreover, phosphorylation of BIN1 at position T348 increases the availability of the SH3 domain for Tau binding, and in AD brains the level of phospho-T348 BIN1 was increased compared to BIN1 [290].

Downregulation of BIN1 levels in neurons results in Tau propagation that is linked to increased endocytosis and blocking endocytosis via dynamin inhibition reduces Tau pathology propagation [291]. The link between BIN1, AD and endocytosis is further highlighted by the role played by BIN1 in regulating the intracellular levels of BACE1, a transmembrane protease responsible for amyloid-beta peptide production, via its endocytosis, endosomal trafficking and lysosomal degradation [292]. In rat primary neurons, the BIN1-Tau protein complexes localize to neuronal soma and dendrites and show a partial colocalization with the actin cytoskeleton (revealed by phalloidin staining) [289]. These data suggest that unraveling the interactions between BIN1 and Tau in link with the regulation of the actin cytoskeleton may advance our understanding of the cytoskeleton alterations observed in Alzheimer’s disease.

## 5. Concluding Remarks and Future Directions

It is clear that the actin cytoskeleton plays a key role in diverse cellular processes including not only control of cell morphology and division, but also processes like protein translation and cellular metabolism. Given the complexity associated with the function of the actin cytoskeleton, the ability to employ powerful molecular genetic approaches becomes crucial to characterize the specific contribution made by each actin cytoskeleton component to the various different cellular processes it may influence. Model eukaryotes that are easily amenable to molecular genetic modification provide the researcher with the capacity to employ molecular genetic approaches to gain novel insights into the function of the actin cytoskeleton at the level of individual molecules.

Here, we have provided four examples illustrating how concurrent studies on both human actin-associated proteins and their link to major diseases, and on functional homologues in budding yeast, have led to a faster and more comprehensive advancement of knowledge. Such parallel studies are important given that actin-associated proteins are implicated in a diverse range of diseases, for example ranging from autoinflammatory disease (e.g., PSTPIP1) to cancer (e.g., BIN1). Even though yeast and humans are somewhat distantly related, studies in budding yeast, and human clinical studies, have resulted in findings that are in general agreement and provided more confidence that the cellular roles of these proteins are in large part conserved through eukaryotic evolution. Each experimental system provides advantages and disadvantages that are unique to that system and hence the experimental systems are complementary. For example, the identification of a potential new human disease protein using population-based approaches such as genome-wide association studies (GWAS) and statistical analysis of patient data, can be complemented by studies on the function of this protein in the yeast model system by investigating e.g., protein-protein interactions and biochemical activities of the yeast homologous protein. Finally, the ability to create, through molecular genetic manipulations, strains of budding yeast that express selected human disease-associated proteins, is opening up new opportunities to screen low molecular weight compound libraries for drugs that selectively target the human protein without the ethical dilemmas associated with human clinical trials or the use of laboratory animals.

## Figures and Tables

**Figure 1 cells-09-00672-f001:**
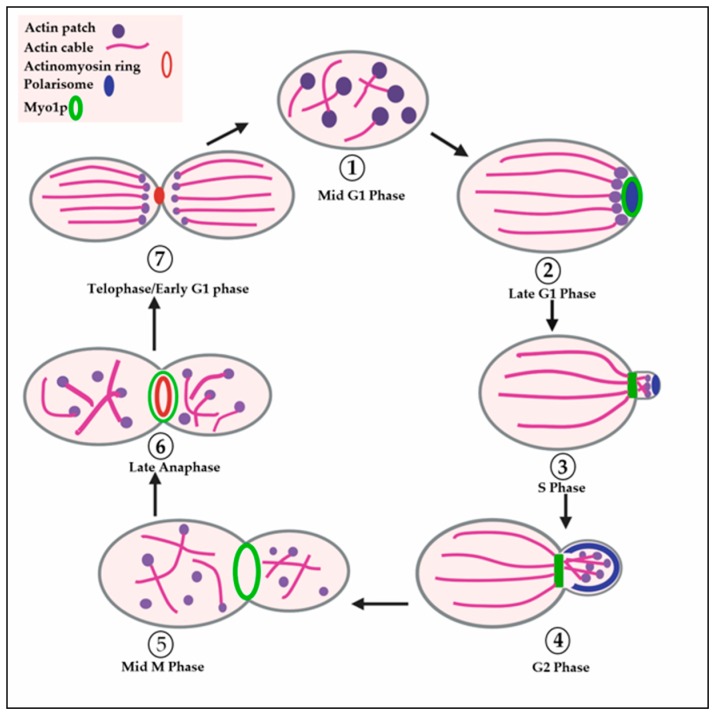
Actin cytoskeleton rearrangement during the cell cycle (in haploid or diploid cells). (**1**) Mid G1 phase: In the cell cycle actin (patches and cables) polarization starts during the shift from mid-G1 (1) to late-G1 phase (2). (**2**) Late G1 phase: Cells choose a new (nascent) bud site and then actin patches start to polarize to this nascent bud site and actin cables orient towards this nascent bud site (N.B. the spatial relationship of the nascent bud site to the previous bud site differs in haploids and diploids). (**3**) S phase: Cortical actin patches cluster at the tip of the bud and actin cables in the mother cell are oriented towards the newly formed bud. (**4**) G2 phase: Actin patches remain polarized to the growing bud but are no longer clustered and become isotropic within the bud while actin cables in the mother cell remain oriented to the growing bud. (**5**) Mid M-phase (mitosis): Actin patches become completely depolarized throughout the mother cell and bud while maintaining localization around the cell cortex and actin cables are randomly oriented. (**6**) Late anaphase: Actin patches and cables are depolarized in the large bud and mother cell and actin is recruited to the Myo1p ring to form an actomyosin ring. (**7**) Telophase/Early G1: Actin patches are polarized and actin cables are oriented to the site of cell division in both the mother cell and bud and contraction of the actomyosin ring results in cytokinesis.

**Figure 2 cells-09-00672-f002:**
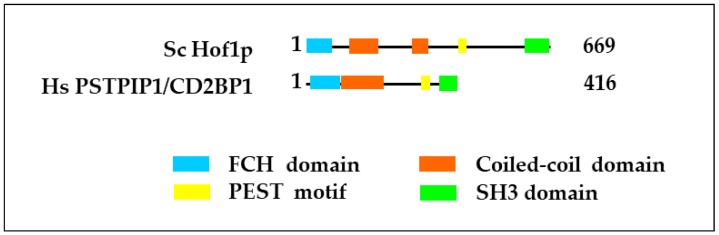
Schematic depicting the domain structure of *Saccharomyces cerevisiae* (*Sc*) Hof1p and *Homo sapien* (*Hs*) PSTPIP1. *FCH domain*: Fes CIP4 Homology domain, *PEST motif*: proline, glutamic acid, serine, threonine-rich motif, *SH3 domain*: Src Homology 3 domain.

**Figure 3 cells-09-00672-f003:**
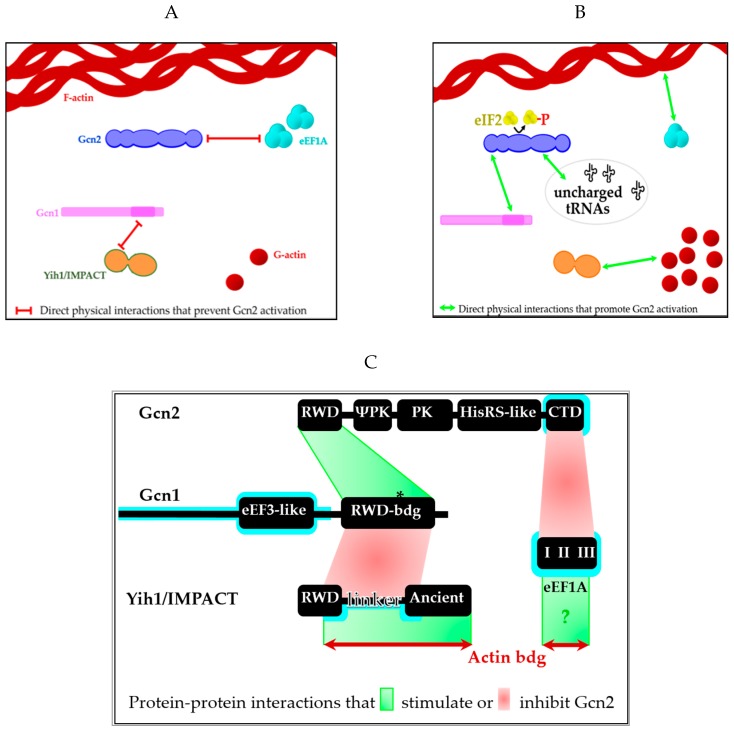
Gcn2 and Yih1p/IMPACT activity is controlled via spatiotemporally constrained rearrangements of the actin cytoskeleton. (**A**) Interactions preventing Gcn2 activation and thus resulting in high rates of protein synthesis. In the current working model, eEF1A binds to Gcn2 to prevent its stimulation. Low G-actin levels dissociate Yih1/IMPACT from actin and Yih1/IMPACT then binds to Gcn1, thereby preventing Gcn1 from activating Gcn2. Current findings suggest that this is due to its inability to promote transfer of the starvation signal (uncharged tRNAs, i.e., tRNA^deacyl^) to Gcn2. Yih1/IMPACT released from actin would allow Yih1/IMPACT to also execute Gcn2-independent functions. Increased *de novo* synthesis of eEF1A, and/or its augmented release from F-actin, enhances eEF1A binding to Gcn2 to prevent its activation. (**B**) Interactions promoting Gcn2 activation and eIF2α phosphorylation by Gcn2 to dampen global protein synthesis and enhance translation of specific mRNAs. Uncharged tRNAs (tRNA^deacyl^) abrogate Gcn2-eEF1A interaction, allowing Gcn2 activation. Enhanced eEF1A interaction with F-actin may also favor dissociation of eEF1A from Gcn2. Actin depolymerization increases the levels of G-actin, which then sequesters Yih1/IMPACT. Sequestration of Yih1/IMPACT allows enhanced Gcn1-Gcn2 interaction, which in turn enhances Gcn2 sensitivity to tRNA^deacyl^. Actin depolymerization leads to increased levels of tRNA^deacyl^ and this further contributes to the activation of Gcn2. Enhanced Gcn2 activity and eIF2α phosphorylation lead to attenuation of global protein synthesis and concomitant enhancement of the expression of Gcn4/ATF4. These major transcriptional regulators adjust the gene expression profile in response to the activating cue that was imposed on the cell. (**C**) Simple schematic showing the domains of Gcn1, Gcn2 and IMPACT and the protein regions known so far to be involved in protein-protein interactions that stimulate or inhibit Gcn2. For simplicity, the ribosome has been omitted in this figure and instead protein regions involved in interactions with the ribosome are shown with a cyan shadow.

**Figure 4 cells-09-00672-f004:**
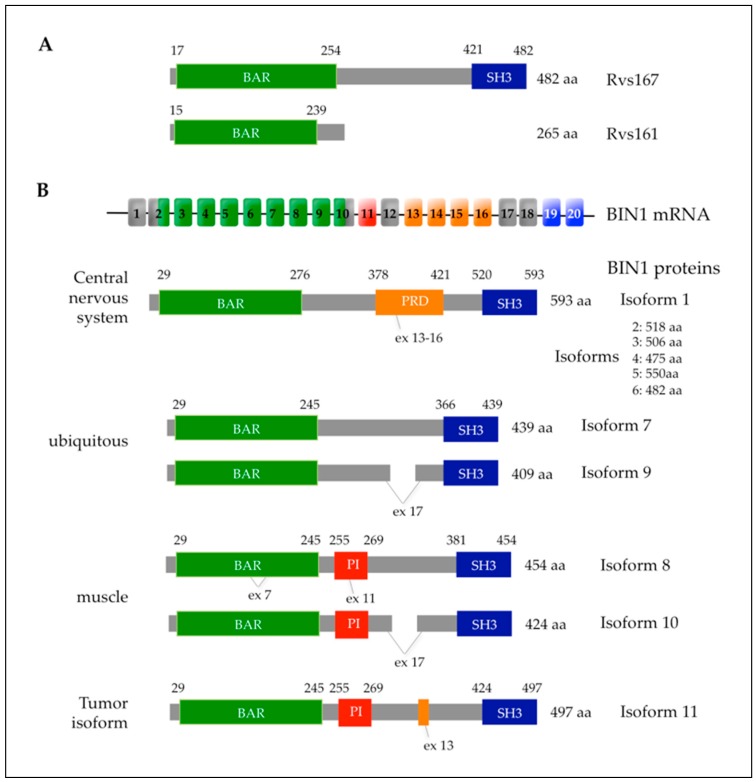
The different domains present in amphiphysins. (**A**) The yeast amphiphysins Rvs161p and Rvs167p. (**B**) The BIN1 mRNA is composed of 20 exons, some being alternatively spliced to give the different isoforms of BIN1 (Uniprot ID O00499). Some isoforms share the same domains. In the central nervous system, there are 6 isoforms termed isoforms 1 to 6, resulting from alternative splicing, only the canonical isoform 1 is shown. The different domains present or not in amphiphysins are: BAR for BIN1/Amphiphysin/Rvs167; SH3, Src homology 3; PRD, proline-rich domain also termed CLAP for Clathrin-Associated Protein Binding domain, encoded by exons 13 to 16 and present in the brain- specific isoforms 1 to 6; PI for Phosphoinositide domain, encoded by exon 11 (previously annotated exon 10) and present in the muscle-specific isoforms 8 and 10, and in the BIN1 tumor isoform 11 (previously termed BIN1 + 12A).

**Table 1 cells-09-00672-t001:** Nomenclature of yeast and mammalian proteins referred to in this article.

Mammals	*S. cerevisiae*
PSTPIP1	Hof1p (cdc15p in *S. pombe*)
BIN1	Rvs167p
IMPACT	Yih1p
GCN2 (in humans also eIF2AK4)	Gcn2p
eEF1A1 and eEF1A2	eEF1a (or Tef1p)

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
