# Peer review of "Yeast as a Model to Understand Actin-Mediated Cellular Functions in Mammals—Illustrated with Four Actin Cytoskeleton Proteins"

_cells, 2020, doi:10.3390/cells9030672_

Round 1
Reviewer 1 Report
The authors have done a good job of addressing the previous concerns. This review should be published.
Author Response
No reviewer 1 comment to address.
Reviewer 2 Report
This review by Akram et al., describes the budding yeast S. cerevisiae as a model system to investigate the function of actin, especially in diseases including cancer. The review is well written and deals with much information about actin-binding proteins of human and yeast. In my opinion, this review is suitable for publication in Cells after the addition of a few revisions as shown below.
The authors described the important interaction between BIN1 and the Myc oncoprotein. Since BIN1 exists both in the cytoplasm and the nucleus, it is supposed that BIN1 associates with nuclear actin. Recently, the information of nuclear actin is rapidly accumulated, but the authors did not mention the nuclear actin and also the possible interaction with BIN1. The authors should provide information about this point.
Author Response
Reviewer 2 Comment:
The authors described the important interaction between BIN1 and the Myc oncoprotein. Since BIN1 exists both in the cytoplasm and the nucleus, it is supposed that BIN1 associates with nuclear actin. Recently, the information of nuclear actin is rapidly accumulated, but the authors did not mention the nuclear actin and also the possible interaction with BIN1. The authors should provide information about this point.
Author Response:
We have revised our manuscript to address the comment of reviewer #2 relating to the subcellular localization of the Bin1-Myc interaction. New text has been added (see underlined text in lines 1168-1173 and 1187). We have also added two new references ([276] and [277]) within the added text and also added these new references to the references section. We have also updated the numbering of the subsequent references in both the text and the references section.